# Control of flagellar gene expression by a chemotaxis receptor-like regulator in pathogenic *Escherichia coli*

Jae-Woo Lee[1,4], Liyun Wang [ID] [1,4], Sarah L Comer[2], Remy Colin[1], Mollie M Gidney[2], Leanid Laganenka[3], Wolf-Dietrich Hardt[3], Maria Hadjifrangiskou [ID] [2] & Victor Sourjik [ID] [1✉]

## Abstract

**Chemoreceptors enable bacteria to modulate their swimming behavior in response to the perceived environmental cues. Despite the large diversity of stimuli sensed by bacterial chemoreceptors, their output to the chemotaxis pathway that controls the flagellar motor typically converges on a few conserved signaling proteins. Here, we characterize a unique chemoreceptor-like protein, Tls, which is found in the B2 phylogroup of *Escherichia coli* that includes many extraintestinal pathogenic (ExPEC) strains. Instead of mediating chemotactic signaling, Tls controls motility by repressing the expression of flagellar genes, and thus cell motility, through sequestration of the transcriptional master activator of flagellar genes, FlhDC. The subcellular localization of Tls, the sequestration of FlhDC, and the repression of flagellar gene expression and motility are all abolished during growth on porous medium, indicating that this regulation may be mechanosensitive. Deletion of *tls* in a uropathogenic *E. coli* strain leads to reduced attachment to the urinary tract cells and an increased migration to and/or proliferation in the murine gut, a pathogen reservoir niche, thus implicating Tls in the regulation of motility during infection.**

**Keywords** Bacteria; Motility; Chemotaxis; Virulence; Mechanosensing; Receptors
**Subject Categories** Cell Adhesion, Polarity & Cytoskeleton; Microbiology, Virology & Host Pathogen Interaction

## Introduction

The majority of motile bacteria possess a chemotaxis signaling system that regulates their motility and enables navigation in chemical and physical gradients (Hazelbauer, 2012; Sourjik and Armitage, 2010; Wadhams and Armitage, 2004). While the core of the chemotaxis pathway is highly conserved, chemoreceptors show large diversity among bacterial species, or even among strains of a single species, depending on their lifestyles (Collins et al, 2014; Ortega et al, 2017;

Wuichet et al, 2007). In most cases, this variability affects the number of chemoreceptors and the ligand specificity of the periplasmic sensory domains, while the overall structural organization of transmembrane chemoreceptors and their coupling to the signaling pathway are retained (Bi and Sourjik, 2018). Some bacteria also contain cytoplasmic chemoreceptors that are assumed to regulate activity of the chemosensory pathway in response to intracellular stimuli (Collins et al, 2014). In the model chemotactic organism, *Escherichia coli*, the reference strain K-12 contains 5 chemoreceptors (Tar, Tsr, Trg, Tap, and Aer) which, together with cytoplasmic chemotaxis proteins, form mixed supramolecular signaling clusters in the cytoplasmic membrane (Bi and Sourjik, 2018; Sourjik, 2004). However, other isolates of *E. coli*, including uropathogenic strains, show variations in their chemoreceptor composition (Borziak et al, 2013; Lane et al, 2006). Although this variation is likely to be associated with differences in their lifestyles, its physiological relevance remains unknown.

The chemotaxis pathway is typically expressed as a part of a larger flagellar regulon which, in *E. coli* and closely related *Enterobacteriaceae*, includes over 60 genes that are hierarchically organized in three different classes of expression (Chevance and Hughes, 2008; Soutourina and Bertin, 2003). At the top of this hierarchy in *E. coli* is the hetero-oligomeric transcriptional regulator consisting of FlhD and FlhC subunits (FlhDC) (Wang et al, 2006), which activates production of flagella. Together with the housekeeping sigma factor, FlhDC mediates transcription of the class 2 flagellar genes required for the assembly of the flagellar motor structure known as the hook-basal body (HBB). Upon HBB completion, class 3 genes encoding the structural filament components, including flagellin FliC and the chemotaxis proteins, are transcribed by the class 2 encoded flagella-specific sigma factor σ[28] (FliA) (Hughes et al, 1993; Karlinsey et al, 2000).

Flagellar biosynthesis consumes several percent of cellular protein budget and has a significant impact on growth fitness (Ni et al, 2020), and it is consequently tightly controlled by various environmental and cellular conditions (Colin et al, 2021; Keegstra et al, 2022; Soutourina and Bertin, 2003). Such control may be even more important in the pathogenic strains, given the high immunogenicity of flagella (Yoon et al, 2012) and the need to migrate to and from infection and reservoir niches within the host. In the best-studied model of *E. coli* and *Salmonella* motility, much of the regulation by environmental factors occurs at the top of the

[1]Max Planck Institute for Terrestrial Microbiology and Center for Synthetic Microbiology (SYNMIKRO), Marburg, Germany. [2]Department of Pathology, Microbiology & Immunology, Vanderbilt University Medical Center, Nashville, TN, USA. [3]Institute of Microbiology, D-BIOL, ETH Zurich, Zurich, Switzerland. [4]These authors contributed equally: Jae-Woo Lee, Liyun Wang. ✉E-mail: victor.sourjik@mpi-marburg.mpg.de

gene expression cascade, targeting transcription of *flhDC* (Chevance and Hughes, 2008; Soutourina and Bertin, 2003), but the activity of FlhDC is also modulated post-translationally by several regulators. These include RflP (YdiV) that prevents FlhDC from DNA binding and targets it for proteolysis (Li et al, 2012; Sporing et al, 2018; Takaya et al, 2012), FliT that also enhances FlhDC proteolysis by the ClpXP protease (Sato et al, 2014; Yamamoto and Kutsukake, 2006), and the chaperone DnaK that promotes correct folding of the FlhDC complex (Takaya et al, 2006).

*E. coli* strains belonging to the B2 phylogroup that includes extraintestinal pathogenic *E. coli* (ExPEC) have a full set of chemotaxis and motility genes, but a reduced set of chemoreceptors, with only Tar, Tsr, and Aer being intact. Whereas Tap is completely lost in these strains, the gene encoding Trg is truncated and still encodes a fraction of the cytoplasmic domain (Borziak et al, 2013). This *trg* version has been previously considered to be a pseudogene, as it cannot perform a regular chemosensory function (Borziak et al, 2013). Here, we demonstrate that this truncated Trg-like signaling protein (Tls) regulates motility in ExPEC. However, rather than controlling the chemotaxis pathway, Tls has been apparently evolutionary reprogrammed to acquire an entirely novel function as a negative regulator of class 2 and class 3 flagellar gene expression, by sequestering the FlhDC complex away from the chromosome. Our results further suggest that Tls controls flagellar biosynthesis in response to the mechanical properties of the growth environment and that this regulation plays an important role in tuning interactions of pathogenic *E. coli* with its host.

## Results

### Trg-like signaling protein (Tls) is conserved among *E. coli* B2 phylogroup strains

We started our analysis by comparing the set of chemoreceptor genes between different phylogroups of *E. coli*. Consistent with the previous reports (Borziak et al, 2013; Lane et al, 2006), genome analysis of the members of *E. coli* B2 phylogroup showed the presence of Tar, Tsr and Aer receptors and the loss of Tap (Appendix Fig. S1), as well as the truncation of the gene encoding Trg (Fig. 1A). The encoded truncated version of Trg, which we tentatively named T̲rg-like s̲ignaling protein (Tls), contains a part of the cytoplasmic signaling domain along with a unique short amino acid sequence MNVRIVLLS at its N-terminus (TlsN; Fig. 1B; Appendix Fig. S2). Tls is highly conserved in the B2 phylogroup, with its N-terminal sequence exhibiting only limited variation among B2 strains (Fig. 1C; Appendix Fig. S2).

### Tls represses flagellar gene expression and motility during *E. coli* growth in liquid

To examine the possible physiological function of Tls, we deleted *tls* gene in *E. coli* B2 strain S13, a septicemia isolate that was previously used to investigate the regulation of motility in pathogenic *E. coli* (Laganenka et al, 2020) and is similar to the prototypic uropathogenic *E. coli* UTI89 (Fig. 1C; Reagents and Tools Table and Appendix Table S1). When tested for spreading in the tryptone broth soft agar (TBSA), a common assay for bacterial motility, S13 Δ*tls* strain behaved similarly to the wild-type strain (Fig. 2A). Since

functional motility as well as the chemotaxis pathway are required for efficient spreading in soft agar, Tls apparently has no major impact on either motility or chemotaxis of *E. coli* S13 under these conditions. A modest reduction of spreading caused by the *tls* deletion might be due to a moderately but reproducibly reduced growth of this strain in tryptone broth (TB) (Appendix Fig. S3A,H).

In contrast to its minor negative effect on spreading in soft agar, the deletion of *tls* had a major positive impact on the motility of *E. coli* S13 grown in liquid TB medium. The wild-type *E. coli* S13 cells swims poorly under these conditions, because the expression of motility genes in this strain is repressed in liquid (Laganenka et al, 2020). However, motility improved upon the deletion of *tls*, resulting in a largely increased fraction of swimmer cells (Fig. 2B,C; Appendix Fig. S3B) and also a more modest but significant increase in the speed of swimming cells (Fig. 2D). The probability of cell reorientations (tumbling bias) also moderately increased in the S13 Δ*tls* strain, measured either in motile cells in the bulk or in those close to the surface of the microscopy chamber (Fig. 2E). Such change of the tumbling bias might provide an alternative explanation for slightly reduced spreading of the S13 Δ*tls* strain in soft agar (Fig. 2A).

We could further show that this inhibition of *E. coli* S13 motility by Tls is due to its negative effect on flagellar gene expression in liquid, with the S13 Δ*tls* mutant exhibiting largely increased activity of the class 3 *fliC* promoter, measured using the corresponding GFP reporter plasmid (Fig. 2F and Reagents and Tools Table). Similar, albeit less pronounced, effect of *tls* deletion on promoter activity could be observed in *E. coli* UTI89 (Fig. 2G), although there was no apparent growth defect or significant effect on swimming under tested growth conditions (Appendix Fig. S3C,D). Confirming its specificity, the impact of *tls* deletion on gene expression could be complemented in a dose-dependent manner by expressing Tls in the S13 Δ*tls* strain from an inducible plasmid vector (Reagents and Tools Table and Appendix Table S1), with high induction levels reducing the *fliC* promoter activity even below the wild-type level (Fig. 2H). Swimming velocity was also reduced in the dose-dependent manner by the Tls expression in the S13 Δ*tls* strain (Appendix Fig. S3E). Finally, the overproduction of Tls in the wild-type S13 strain further reduced the *fliC* promoter activity (Appendix Fig. S3F).

Similar dose-dependent inhibitory effects of Tls expression on *fliC* promoter activity and on cell motility were observed in *E. coli* MG1655 (Fig. EV1A–C). Since MG1655 has no native *tls* gene but instead encodes Trg, these experiments are similar to the expression of Tls in the S13 Δ*tls* strain (Fig. 2H; Appendix Fig. S3E,F). In addition, they demonstrate that the Tls function as a negative regulator of flagellar gene expression in *E. coli* does not necessarily require other proteins that are specific for the B2 phylogroup. In general, effects of Tls on motility were weaker than on gene regulation, both in MG1655 and in S13, which might be due to the limited sensitivity of *E. coli* motility to flagellar gene expression, particularly at high expression levels (Lisevich et al, 2025). The expression of Tls had no negative impact on the growth of either *E. coli* MG1655 or S13 even at high levels of induction, ruling out indirect growth-dependent regulation as a possible cause of changes in flagellar gene expression (Appendix Fig. S3G,H). Finally, we used a microfluidics-based chemotaxis assay (Colin et al, 2019; Colin et al, 2014) to test possible impact of Tls on the chemotactic behavior of *E. coli*. Although the expression of Tls in

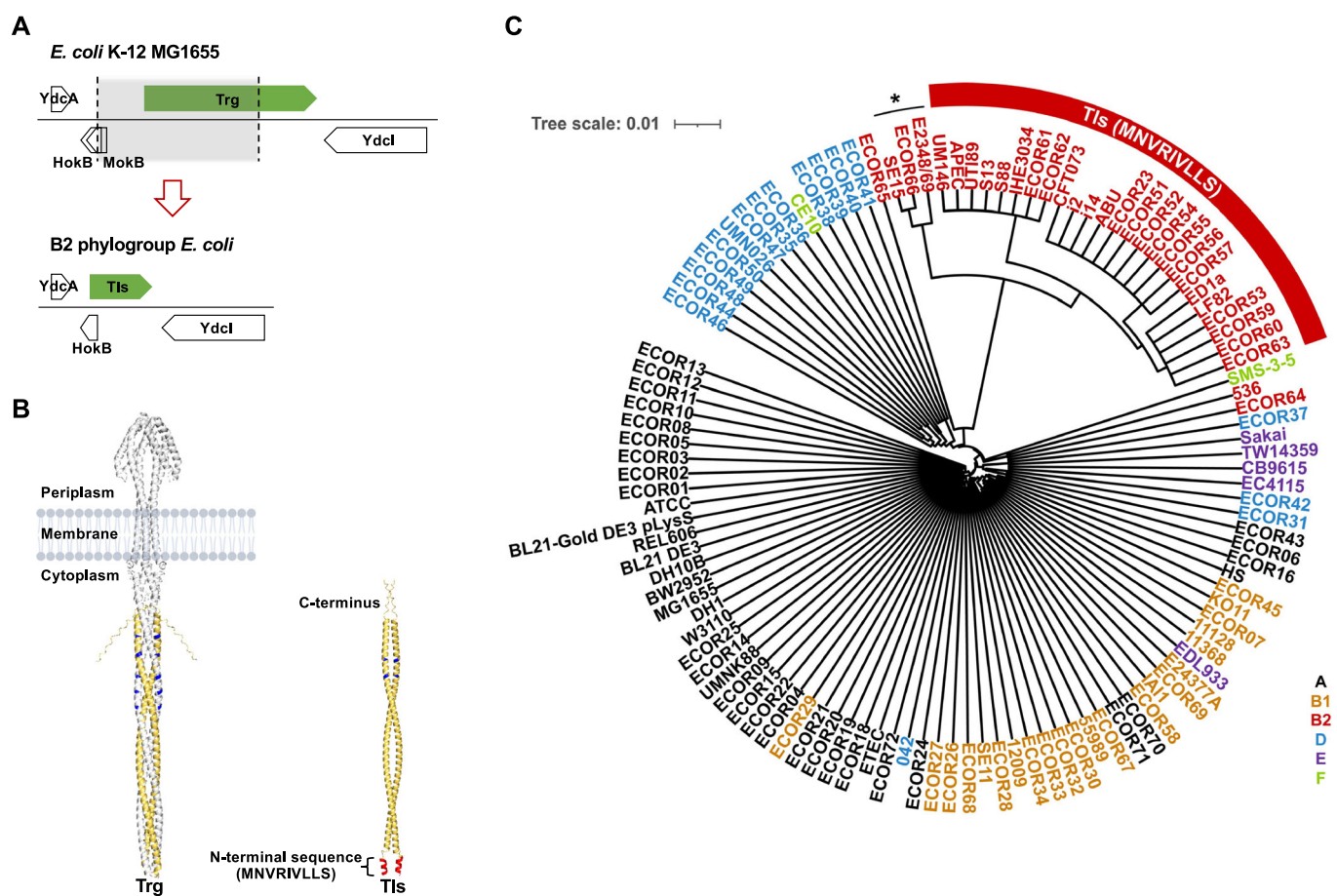

**Figure 1. Identification of Trg-like signaling protein (Tls).**

(**A**) Comparison between the genomic region containing *trg* gene in *E. coli* MG1655 and the corresponding region in the B2 phylogroup of *E. coli*, represented by UTI89, based on the NCBI and MiST4.0. Genes are labeled with the names of their products, including truncated Trg-like signaling protein (Tls) instead of Trg in the B2 phylogroup. Besides the truncation of *trg*, deletion of the shaded genomic region in the B2 phylogroup resulted in the loss of the divergently transcribed gene *mokB* that is unrelated to motility. (**B**) Structural models of the full-length Trg (left) and of Tls (right), constructed using AlphaFold3. The C-terminal part common between Trg and Tls is marked in yellow. A Tls-specific short N-terminal amino acid sequence (TlsN) in marked in red. Methylation sites of Trg and corresponding residues in Tls are labeled in blue. Trg is shown in its functional dimeric form. Although the oligomeric state of Tls is unknown, the prediction for a dimeric structure is shown by analogy with Trg. (**C**) Presence of Tls in the completely sequenced *E. coli* genomes, isolates from the ECOR collection, and *E. coli* S13. The phylogenetic tree is based on the *trg* nucleotide sequences and strains are colored according to previously established phylotypes. Tree scale is 0.01 substitutions per site. Tls with the conserved TlsN sequence MNVRIVLLS is present in the majority of the B2 phylogroup *E. coli* strains. ECOR65 has intact *trg* nucleotide sequence, and *E. coli* strain 536 and ECOR64 have frameshift in *trg* nucleotide sequence. An asterisk denotes Tls variant that carries the N-terminal sequence MRFSQFNHSLLS (TlsN*).

MG1655 moderately reduced the chemotactic drift velocity measured in this assay (Fig. EV1D) by lowering the cell swimming velocity, the chemotactic bias (which is a more specific measure of chemotaxis) was not affected by the expression of Tls (Fig. EV1E).

## Tls controls class 2 flagellar genes

To better understand the nature of the Tls-dependent regulation, we first determined its impact on the activity of *flhD* (class 1), *fliA* (class 2), and *fliC* (class 3) promoters. Whereas the *flhD* promoter activity was not altered, the reporter expression from *fliA* and *fliC* promoters dramatically increased in S13 Δ*tls* compared to the wild-type strain (Fig. 3A). Consistently, expression of Tls in MG1655 inhibited activities of the class 2 and class 3 promoters but not of the class 1 promoter (Fig. 3B). We thus performed the whole proteome analyses of the impact of *tls* deletion in *E. coli* S13, to

examine derepression of Tls-dependent genes. Complementary to that, we also analyzed their repression upon expression of Tls in MG1655. Consistent with the flagellar gene reporter assays, proteomic analyses demonstrated an increase in the abundance of nearly all of the class 2 and class 3 flagellar proteins upon deletion of *tls* in *E. coli* S13 (Fig. 3C), and their decrease upon expression of Tls in MG1655 (Fig. 3D). The regulation by Tls was largely limited to flagellar and chemotaxis proteins, which formed the main functional group that was derepressed in S13 Δ*tls* and/or repressed in MG1655 expressing Tls, according to the STRING database (Szklarczyk et al, 2023) (Fig. EV2; Appendix Tables S2–S5; Dataset EV1).

Interestingly, and in contrast to other flagellar proteins that were downregulated, the abundance of FlhD and FlhC even increased upon expression of Tls (Fig. 3D). This regulation indicated that Tls might protect FlhDC against degradation while inhibiting its

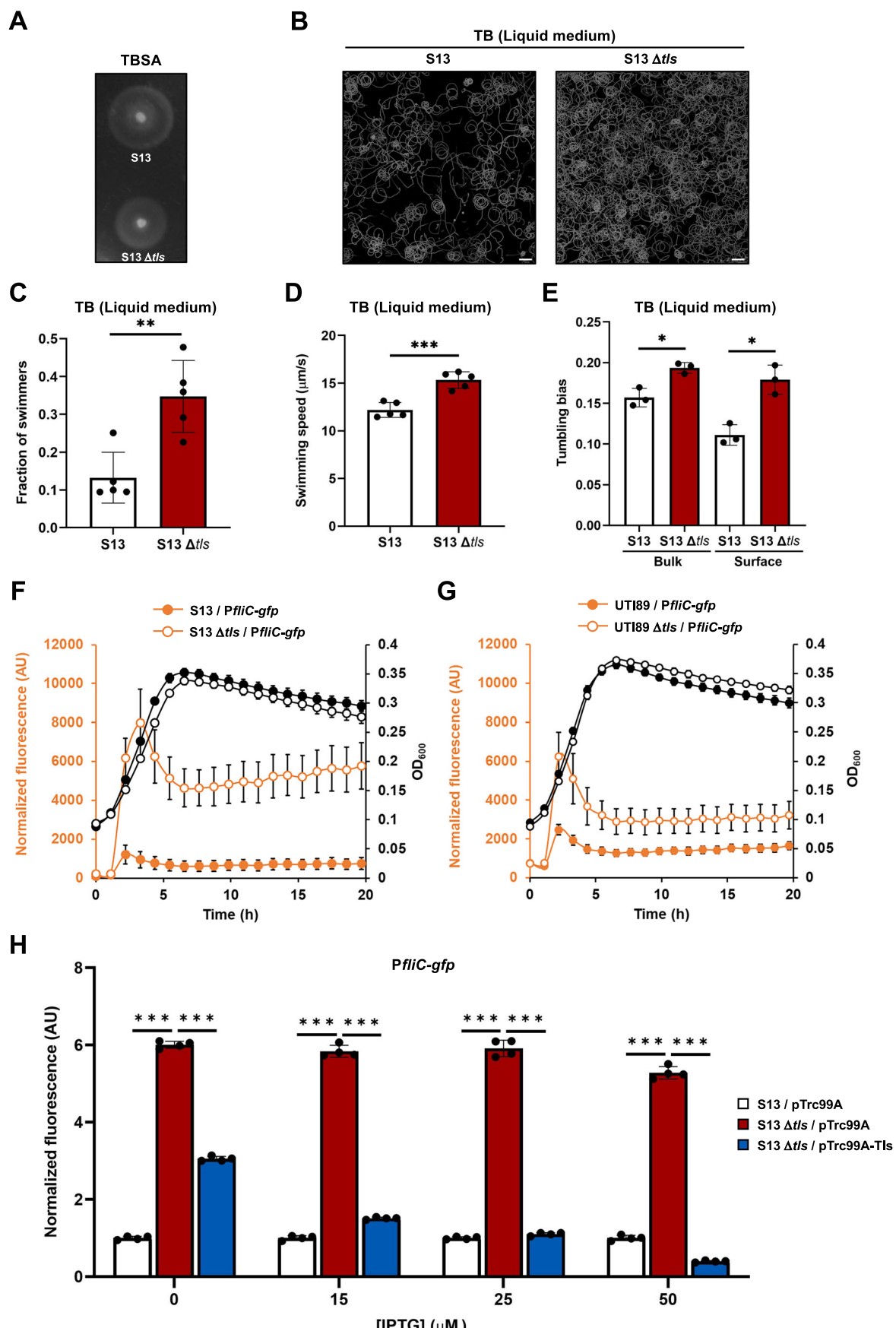

**Figure 2. Negative effect of Tls on motility in liquid medium.**

(A) Motility- and chemotaxis-dependent spreading of *E. coli* S13 and its Δ*tls* mutant in 0.27% TB soft agar (TBSA). Representative image is shown from three biological replicates. (B) Swimming trajectories of *E. coli* S13 and S13 Δ*tls* cells close to the glass surface in the microscopy chamber, shown as their projected intensities (white curves on dark background) during 20-s long movies. Representative images are shown from three biological replicates. Scale bars, 50 μm. (C, D) Fractions of swimmers (C) and swimming speed (D) measured in the bulk of the bacterial suspension (middle of the microscopy chamber) via Differential Dynamic Microscopy (DDM) for *E. coli* S13 and S13 Δ*tls* cells. Values represent the means and standard deviations of five biological replicates. (E) Tumbling bias in the bulk or close to the surface, measured via particle tracking as the averaged fraction of time spent tumbling over 2913 (S13 middle), 4230 (S13 top), 5225 (S13 Δ*tls* middle) and 5281 (S13 Δ*tls* top) swimmer trajectories in three biological replicates. (F, G) Bacterial growth ($OD_{600}$) and fluorescence levels of P*fliC-gfp* reporter normalized by $OD_{600}$ in *E. coli* S13 and its Δ*tls* mutant (F), and in *E. coli* UTI89 and its Δ*tls* mutant (G). Measurements were performed in a plate reader. Values represent the means and standard deviations of a minimum of three biological replicates. (H) Fluorescence levels of P*fliC-gfp* normalized by $OD_{600}$ in *E. coli* S13 and its Δ*tls* mutant, both carrying an empty vector pTrc99A, and in S13 Δ*tls* complemented by expression of Tls from pTrc99A-Tls plasmid induced by indicated concentrations of IPTG. Measurements were performed in the log phase of growth in a plate reader. Values represent the means and standard deviations of four biological replicates, in each case normalized to the reporter activity in the reference strain (S13/pTrc99A). Statistical significance was determined using unpaired two-tailed Student's *t* test. The *P* values are denoted as ns ($P > 0.05$), * ($P < 0.05$), ** ($P < 0.005$), *** ($P < 0.001$). (**$P = 0.0032$ (C); ***$P = 0.0003$ (D); *$P = 0.0085$, *$P = 0.0058$ (E); *P* values from left to right: ***$P = 6.67E-11$, ***$P = 2.69E-09$, ***$P = 1.75E-09$, ***$P = 2.58E-09$, ***$P = 7.41E-09$, ***$P = 8.68E-09$, ***$P = 5.20E-09$, ***$P = 1.51E-09$ (H). Source data are available online for this figure.

activity (see below). Besides FlhDC, a number of proteins showed changes in expression that were opposite to the majority of flagellar proteins. Several of these proteins belong to the general stress response regulon, which is known to be counter-regulated with the flagellar regulon (Pesavento et al, 2008). The upregulation of these proteins upon Tls expression might thus be an indirect consequence of the repression of FlhDC activity. Indeed, deletion of *flhDC* is known to elicit changes in expression of multiple genes beyond the flagellar regulon (Pruss et al, 2001). Consistent with that, most of the Tls-induced changes in protein expression were no longer observed in the MG1655 Δ*flhC* background (Appendix Fig. S4, Appendix Tables S6 and S7; Dataset EV1). We conclude that Tls acts as a specific negative regulator of class 2 flagellar gene expression, likely through the inhibition of FlhDC activity, which then indirectly affects the expression of class 3 flagellar genes as well as of other *E. coli* genes, including the stress response regulon.

## Gene regulation by Tls and its subcellular localization depend on the growth environment

Since the inhibitory effect of Tls on motility was only observed when bacterial cells are grown in the liquid medium, but not in the soft agar (Fig. 2; Appendix Fig. S5), we assumed that the Tls-mediated inhibition might be related to the mechanosensitive regulation of flagellar gene expression that was previously reported in *E. coli* S13 (Laganenka et al, 2020). To test this, we compared the impact of Tls on *fliC* promoter activity in *E. coli* cells grown either in the liquid TB medium and on the surface of semi-solid (0.5%) TB agar, respectively. Whereas the presence of Tls strongly reduced reporter activity in the wild-type S13 compared to S13 Δ*tls* when cells were grown in the liquid medium (Fig. 4A), reporter activity in cells grown on the semi-solid medium was comparably high in both strains (Fig. 4B). Similar result was observed when Tls was expressed in MG1655 (Fig. 4C,D).

RflP (alternatively called YdiV), a negative regulator of FlhDC stability (Li et al, 2012; Wada et al, 2012), was previously shown to play a role in mechanosensing (Laganenka et al, 2020). We thus tested whether the effect of Tls on flagellar genes may depend on RflP. Although the activity of *fliC* promoter was expectedly elevated in S13 Δ*rflP*, the repressive effect of Tls expression was still observed (Appendix Fig. S6A). Importantly, the Tls-dependent regulation remained surface-dependent even in the absence of *rflP*, with repression of flagellar genes only occurring during growth in

liquid (Appendix Fig. S6B), but not on the semi-solid medium (Appendix Fig. S6C). We conclude that Tls regulates flagellar expression together with, but independently of RflP, with the latter being important to lower the basal gene expression as already suggested before (Laganenka et al, 2020).

To obtain further insights into the possible mechanism of Tls-dependent regulation, we imaged the localization of Tls fused to superfolder GFP (sfGFP). Since our previous results demonstrated that Tls is functional in MG1655, fluorescence microscopy experiments were primarily performed in this background. We observed that a large fraction of Tls-sfGFP localized to distinct foci near cell poles when bacterial cells were grown in the liquid medium (Fig. 4E). In contrast, Tls-sfGFP was mainly dispersed in the cytoplasm in surface-grown cells (Fig. 4F), although the intact fusion protein was present at similar levels in these cells (Appendix Fig. S7A). Polar localization of Tls-sfGFP was also strongly reduced in MG1655 cells grown in soft agar (Appendix Fig. S7B,C). Upon reversal of the growth conditions from liquid to surface or vice versa, the localization pattern of Tls-sfGFP also reversed within 30–90 min (Fig. 4G; Appendix Fig. S7D,E).

Tls contains a fragment of Trg, which is a part of the chemosensory clusters that are formed through interactions between receptors and other chemotaxis proteins, and also primarily localize to *E. coli* cell poles (Sourjik, 2004). We thus tested whether the polar localization and function of Tls may depend on these clusters. However, Tls still mediated surface-dependent regulation of class 2 gene expression in Δ*fliA* (Fig. EV3A–D) and changed its subcellular localization in Δ*fliA* (Fig. EV3E,F) and in Δ*flhC* backgrounds (Fig. EV3G,H). Because these strains have strongly reduced levels of chemotaxis proteins, which are encoded by class 3 genes, we conclude that the Tls-mediated repression of flagellar gene expression in liquid and its localization do not require the chemotaxis signaling system (or other proteins encoded by class 3 genes). This was further confirmed by expressing Tls in another *E. coli* K12 strain (RP437) and in its mutants lacking either the chemoreceptor or *cheR* genes. In both cases, the repression of *fliC* promoter activity upon expression of Tls was similar to the parental strain (Fig. EV3I–K).

## N-terminal region is crucial for subcellular localization and gene repression by Tls

Given that Tls consists of a fragment of the cytoplasmic signaling domain of Trg and a short Tls-specific amino acid sequence TlsN at

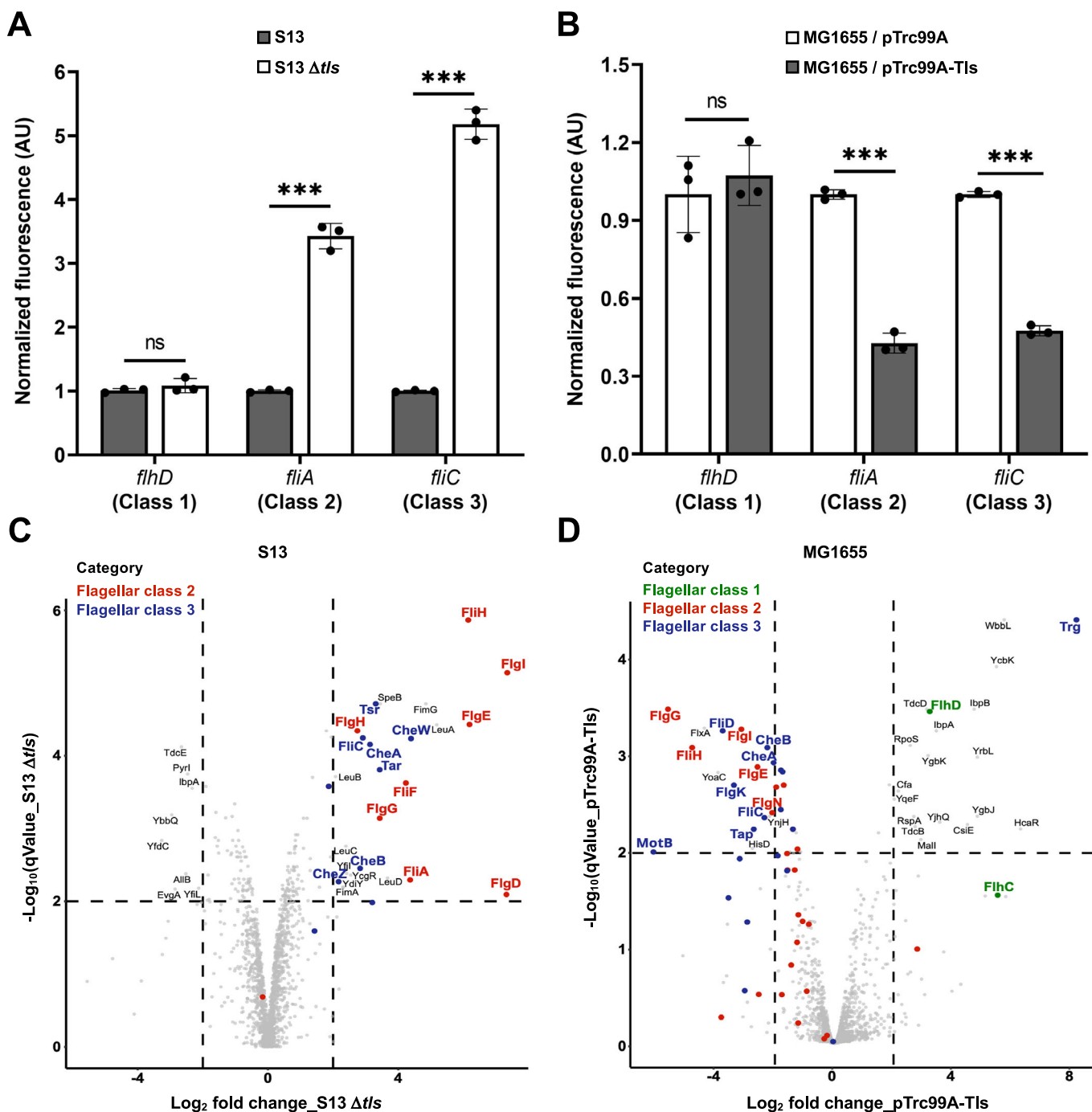

**Figure 3. Downregulation of class 2 and class 3 flagellar gene expression by Tls.**

(A, B) Fluorescence levels of P*flhD-gfp* (class 1), P*fliA-gfp* (class 2), and P*fliC-gfp* (class 3) normalized by OD$_{600}$ in *E. coli* S13 and its Δ*tls* mutant (A), or in MG1655 carrying either empty vector pTrc99A or pTrc99A-Tls induced with 15 μM IPTG (B). Measurements were performed in the log phase of growth in a plate reader. Values represent the means and standard deviations of a minimum of three biological replicates, in each case normalized to the reporter activity in the respective reference strain (S13 or MG1655/pTrc99A). Statistical significance was determined using an unpaired two-tailed Student's *t* test. The *P* values are denoted as ns (*P* > 0.05), * (*P* < 0.05), ** (*P* < 0.005), *** (*P* < 0.001). (*P* values from left to right: ns = 0.32, ***P = 3.01E-05, ***P = 6.82E-06 (A); ns = 0.54, ***P = 2.03E-05, ***P = 2.15E-06 (B)). (C, D) Flagellar proteins detected in the whole proteome analysis of S13 Δ*tls* compared to S13 (C) or MG1655 expressing Tls from pTrc99A-Tls at 15 μM IPTG induction compared to MG1655/pTrc99A (D). Data are from three biological replicates. Statistical significance was determined using moderated *t*-statistics from the empirical Bayes procedure LIMMA (Kammers et al, 2015) to calculate *P* values, followed by adjustment to obtain *q* values based on the Benjamini–Hochberg method (Benjamini and Hochberg (1995)). Significantly increased or decreased flagellar and chemotaxis proteins are colored according to their gene regulatory class, as indicated. For (C), the *E. coli* UTI89 protein annotation database was used as a reference. For (D), although below the statistical threshold, class 1 flagellar protein FlhC was labeled, too. Source data are available online for this figure.

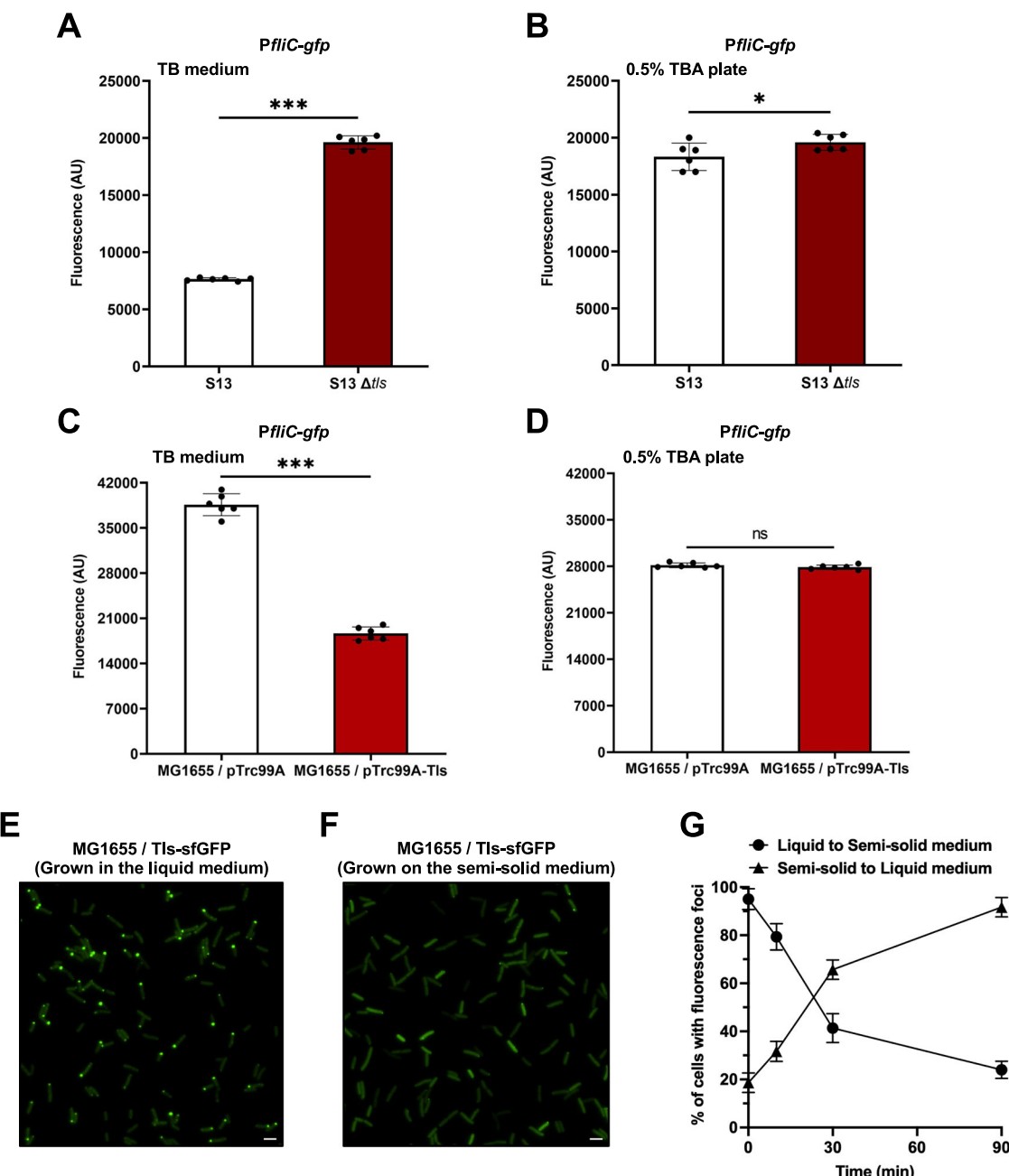

**Figure 4. Surface-sensitive regulation of flagellar gene expression by Tls.**

(A–D) Fluorescence levels of P*fliC-gfp* normalized by $OD_{600}$ in *E. coli* S13 or S13 Δ*tls* cells (**A, B**) or MG1655 cells carrying either empty pTrc99A vector or pTrc99A-Tls expression plasmid induced with 15 μM IPTG (**C, D**). Cells were grown either in liquid TB medium (**A, C**) or on the surface of 0.5% TB agar (TBA) plate (**B, D**). Measurements were performed in the log phase of growth in a plate reader. Values represent the means and standard deviations of a minimum of three biological replicates. Statistical significance was determined using unpaired two-tailed Student's *t* test. The *P* values are denoted as ns (*P* > 0.05), * (*P* < 0.05), ** (*P* < 0.005), *** (*P* < 0.001). (***$P$ = 3.22E-13 (**A**); *$P$ = 0.048 (**B**); ***$P$ = 2.71E-10 (**C**); ns = 0.17 (**D**)). (**E, F**) Subcellular localization of Tls-sfGFP fusion protein in MG1655 cells carrying pTrc99A-Tls-sfGFP and grown either in liquid TB medium (**E**) or on 0.5% TBA plate (**F**). Tls-sfGFP expression was induced by 15 μM IPTG. Representative images are shown from three biological replicates. Scale bars, 3 μm. (**G**) Time-dependent changes in subcellular localization of Tls-sfGFP upon reversal of growth conditions. MG1655 cells expressing Tls-sfGFP were grown either in liquid TB medium or on the 0.5% TBA plate for 3.5 h and transferred to the opposite growth condition for the indicated period of time. Quantification of the number of cells showing fluorescence foci formed by Tls-sfGFP was based on microscopy images in Appendix Fig. S7D,E. For each image, foci were counted in 100 cells from three different images, originating from three biological replicates. Values represent the means and standard deviations. Source data are available online for this figure.

its N-terminus (Fig. 1B), we asked how these parts may contribute to its functionality. Although neither the full-length Trg nor TlsN alone showed any repression of flagellar gene expression in MG1655 (Appendix Fig. S8A,B), fusing TlsN to the cytoplasmic fragment of Trg that corresponds (and is nearly identical) to Tls restored the regulation of flagellar gene expression (Appendix Fig. S8B). Similarly, functional hybrids were obtained by fusing TlsN to the corresponding cytoplasmic fragments of other *E. coli* chemoreceptors (Tar, Tsr, Tap, and Aer). We thus hypothesized that the N-terminal amino acid fragment may be the primary determinant of the observed regulation of gene expression. Since TlsN is apparently non-functional on its own, we fused it to the red fluorescent protein mCherry (which does not interfere with fluorescence measurements of the GFP reporter activity) (Appendix Fig. S8C,D). Expression of the TlsN-mCherry fusion construct indeed resulted in a partial but significant repression of flagellar genes in MG1655 and S13 Δ*tls* backgrounds (Appendix Fig. S8C,D), although this reduction was apparently not sufficient to significantly inhibit *E. coli* motility (Appendix Fig. S8E,F). In contrast, the control construct with a stretch of glycine residues showed no repression of gene expression.

We observed that not only the full-length Tls fused to mCherry (Tls-mCherry) but also TlsN-mCherry exhibited polar localization in liquid-grown cells (Fig. 5A), whereas no localization was observed for the inactive control construct carrying a stretch of glycine residues. Moreover, the repressive effect of TlsN-mCherry and its polar localization were abolished when the positively charged arginine (R4) was replaced by the negatively charged aspartate, whereas a conservative replacement with lysine retained both functionalities (Fig. 5A,B). Consistently, the fusion of TlsN variant (TlsN*), present in *E. coli* isolate E2348/69 that does not have this arginine residue, showed neither polar localization nor gene repression (Fig. 5A,B). Finally, the inhibitory effects of TlsN-mCherry on gene expression and its localization were both absent when bacterial cells were grown on the semi-solid medium (Fig. 5A,B).

Despite the importance of its N-terminal sequence, the remaining Trg-like fragment significantly enhances the regulatory activity of Tls (Fig. 5B). One potential contribution of the fragment to polar localization of Tls could be its ability to dimerize, which is common for the cytoplasmic fragments of bacterial chemoreceptors (Fig. 1B). In this context, we noted a recent report that mCherry might possess a residual ability to dimerize in *E. coli* (Fraikin et al, 2025). We therefore compared the TlsN fusion to the monomeric version of sfGFP (used in all our experiments) with its fusions to the sfGFP variants where the dimerization-favoring mutations N149Y or R206A (Roberts et al, 2016; Zacharias et al, 2002) were introduced. Different from TlsN-mCherry, the monomeric TlsN-sfGFP showed nearly no cases of cellular localization (Fig. 5C; Appendix Fig. S8G). However, both dimeric versions of this fusion, TlsN-sfGFP$^{N149Y}$ and TlsN-sfGFP$^{R206A}$, exhibited pronounced localization during growth in liquid but not on the semi-solid medium, supporting the hypothesis that dimerization may be important for Tls localization.

## Tls forms dynamic structures sequestering FlhDC

We next studied the cellular dynamics of foci formed by Tls-sfGFP by performing fluorescence recovery after photobleaching (FRAP)

microscopy. We observed that fluorescence in regions of the cell containing Tls foci recovered within ~10 s after bleaching by a high-intensity laser light (Appendix Fig. S9). Such recovery is much faster than that observed for chemoreceptor clusters (Schulmeister et al, 2008) or for unspecific protein aggregates (Winkler et al, 2010), suggesting that foci formed by Tls are highly dynamic structures. Notably, similarly rapid dynamics of fluorescence recovery were previously described as a characteristic of biomolecular condensates (Shin and Brangwynne, 2017).

Since Tls represses the expression of class 2 flagellar genes that are controlled by FlhDC, whereas the abundance of FlhD and FlhC proteins themselves even increases in the presence of Tls (Fig. 3D), we hypothesized that Tls might interfere with the activity of FlhDC via its recruitment away from the DNA. Such sequestration could prevent FlhDC from activating the expression of flagellar genes but also from proteolytic degradation, as has been described for the downstream regulator of flagellar gene expression FliA (Barembruch and Hengge, 2007; Rudenko et al, 2019). To validate this hypothesis, we studied the impact of Tls expression in *E. coli* MG1655 on the subcellular localization of FlhDC, expressed from a monocistronic construct with untagged FlhD and FlhC fused to sfGFP at its C-terminus. Although this FlhDC fusion cannot complement the motility defect in Δ*flhC* strain (Appendix Fig. S10), probably due to the interference of sfGFP with the binding to RNA polymerase, it localizes to the nucleoid and the cytoplasm of MG1655 cells, as expected for a DNA-binding transcription factor (Fig. 6A). We further observed that FlhDC-sfGFP partly relocated towards cell poles in the presence of Tls when cells were grown in the liquid medium (Fig. 6A,C). This sequestration was highly significant and observed in the majority of cells, although its extent was less compared to the inhibition of gene expression at the same level of Tls induction (Fig. EV1A), possibly because the amount of Tls in this experiment was not sufficient to sequester all fusion proteins or due to the reduced ability of FlhDC-sfGFP to bind Tls compared to the untagged FlhDC. It is also possible that a large fraction of the Tls-sequestered FlhDC complexes remains in the cytoplasm (see "Discussion"). The sequestration did not occur when Tls-expressing cells were grown on the surface, suggesting the specificity of the observed effect and consistent with the dependence of gene regulation on growth conditions (Fig. 6B,C). We additionally generated an FlhDC variant (FlhC$^{K42E\ K45E}$) that is deficient in DNA binding (Li et al, 2012). As expected, this variant was distributed uniformly across the cytoplasm in the absence of Tls, and it exhibited an even more pronounced polar localization in the presence of Tls (Fig. 6D,E), likely because the sequestration of FlhC$^{K42E\ K45E}$-sfGFP by Tls no longer competes with the FlhDC-sfGFP binding to DNA. In contrast, the expression of Trg did not promote localization even for this FlhDC variant. These results support our hypothesis that Tls can, directly or indirectly, sequester FlhDC away from DNA and thereby prevent it from activating class 2 flagellar gene expression.

The sequestration of FlhDC by Tls should lead to the immediate proximity of both proteins that can be detected using Förster (fluorescence) resonance energy transfer (FRET) between their fusions to the donor and acceptor fluorophores. We thus co-expressed FlhDC-sfGFP (donor fluorophore) with either Tls-mCherry or only mCherry (acceptor fluorophore) and used acceptor photobleaching to detect FRET (Li and Sourjik, 2011; Yadavalli et al, 2020). In liquid-grown bacteria, a highly significant FRET signal

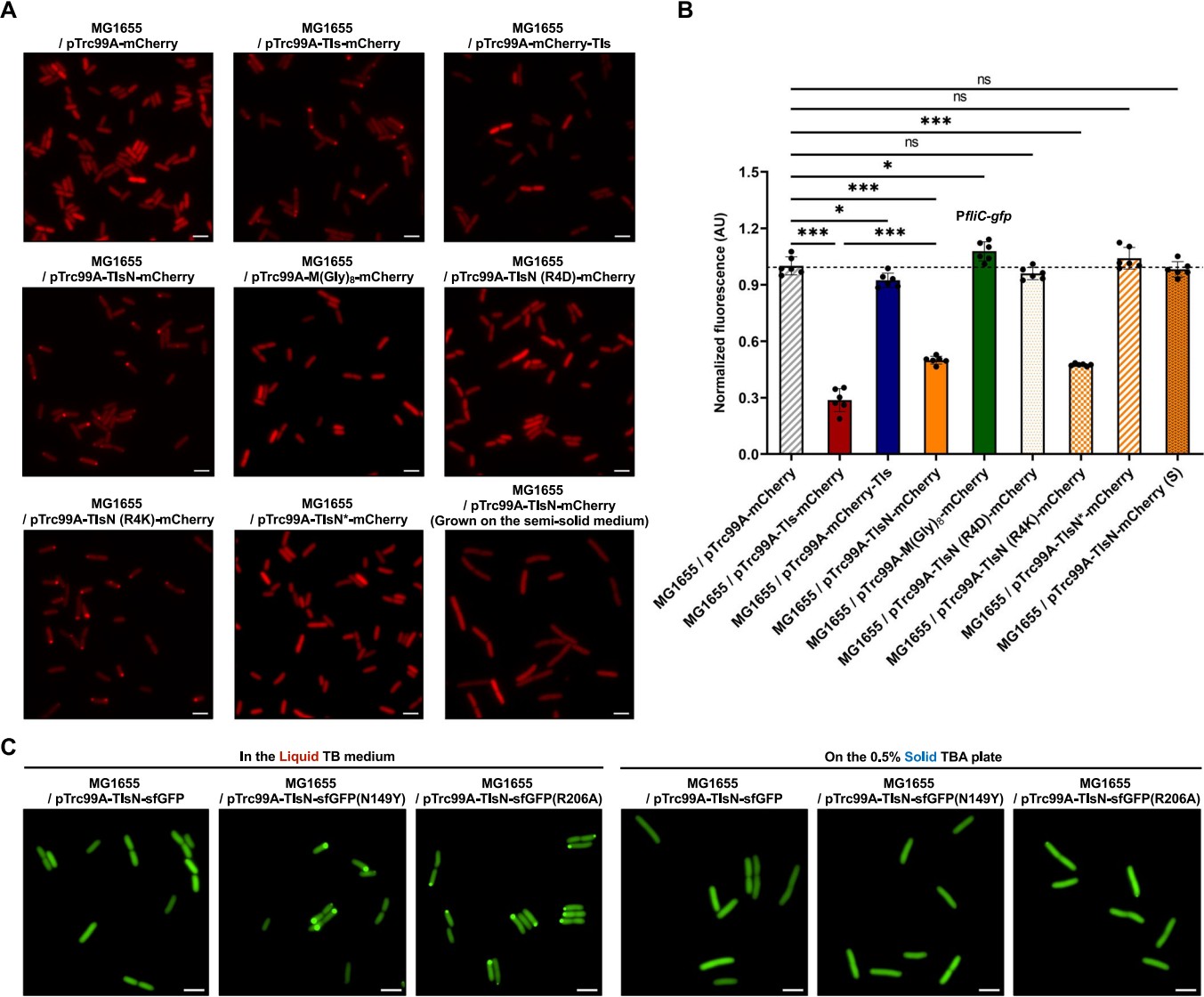

**Figure 5. Importance of the N-terminal Tls sequence for localization and activity.**

(A) Localization of mCherry, C-terminal or N-terminal mCherry fusions to full-length Tls, or C-terminal fusions to indicated variants of the N-terminal Tls sequence (MNVRIVLLS = TlsN), expressed from pTrc99A-derived constructs in MG1655 cells at 15 μM IPTG induction. TlsN* denotes TlsN sequence variant MRFSQFNHSLLS. Representative images are shown from three biological replicates. Scale bars, 2.5 μm. (B) Corresponding fluorescence levels of *PfliC-gfp* reporter in these strains normalized by OD600. Cells were grown in liquid TB in a plate reader or on the semi-solid TB medium (0.5% TBA plate; S) for 3.5 h and fluorescence was measured in a plate reader. Values represent the means and standard deviations of a minimum of three biological replicates, in each case normalized to the reporter activity in the reference strain (MG1655/pTrc99A-mCherry). Statistical significance was determined using unpaired two-tailed Student's *t* test. The *P* values are denoted as ns (*P* > 0.05), * (*P* < 0.05), ** (*P* < 0.005), *** (*P* < 0.001). (The *P* value of Tls-mCherry vs TlsN-mCherry = 1.11E-05; *P* values from left to right: ***P = 6.63E-10, *P = 0.012, ***P = 4.18E-10, *P = 0.022, ns = 0.11, ***P = 1.22E-10, ns = 0.21, ns = 0.49). (C) Subcellular localization of TlsN fusion to either monomeric sfGFP or its dimeric sfGFP(N149Y) or sfGFP (R206A) variants in MG1655 cells grown either in liquid TB medium or on the surface of 0.5% TB agar (TBA) plate. Expression was induced with 15 μM IPTG in liquid TB and with 60 μM IPTG on 0.5% TBA plate. Representative images are shown from three biological replicates. Scale bars, 2.5 μm. Quantification of the number of cells showing fluorescence foci is shown in Appendix Fig. S8G. Source data are available online for this figure.

could be observed for the FlhDC-sfGFP/Tls-mCherry but not for the control FlhDC-sfGFP/mCherry pair (Fig. 6F). In contrast, much weaker FRET signal was observed when cells were grown on solid medium. These FRET results were consistent with the localization of Tls-mCherry and FlhDC-sfGFP (Figs. 6G and EV4A,B): In liquid-grown cells, Tls-mCherry formed highly pronounced foci that apparently recruited FlhDC-sfGFP, to even greater extent than the untagged Tls (Fig. 6A). In surface-grown cells, localization of Tls-

mCherry was strongly reduced, and no FlhDC-sfGFP sequestration could be observed. Such dependence of complex formation between Tls-mCherry and FlhDC-sfGFP on growth conditions suggests that it is either modulated or mediated by other cellular component(s) that remain to be identified.

FRET experiments in the S13 background yielded similar results, confirming the complex formation between FlhDC-sfGFP and Tls-mCherry in the liquid-grown but not in the surface-grown cells

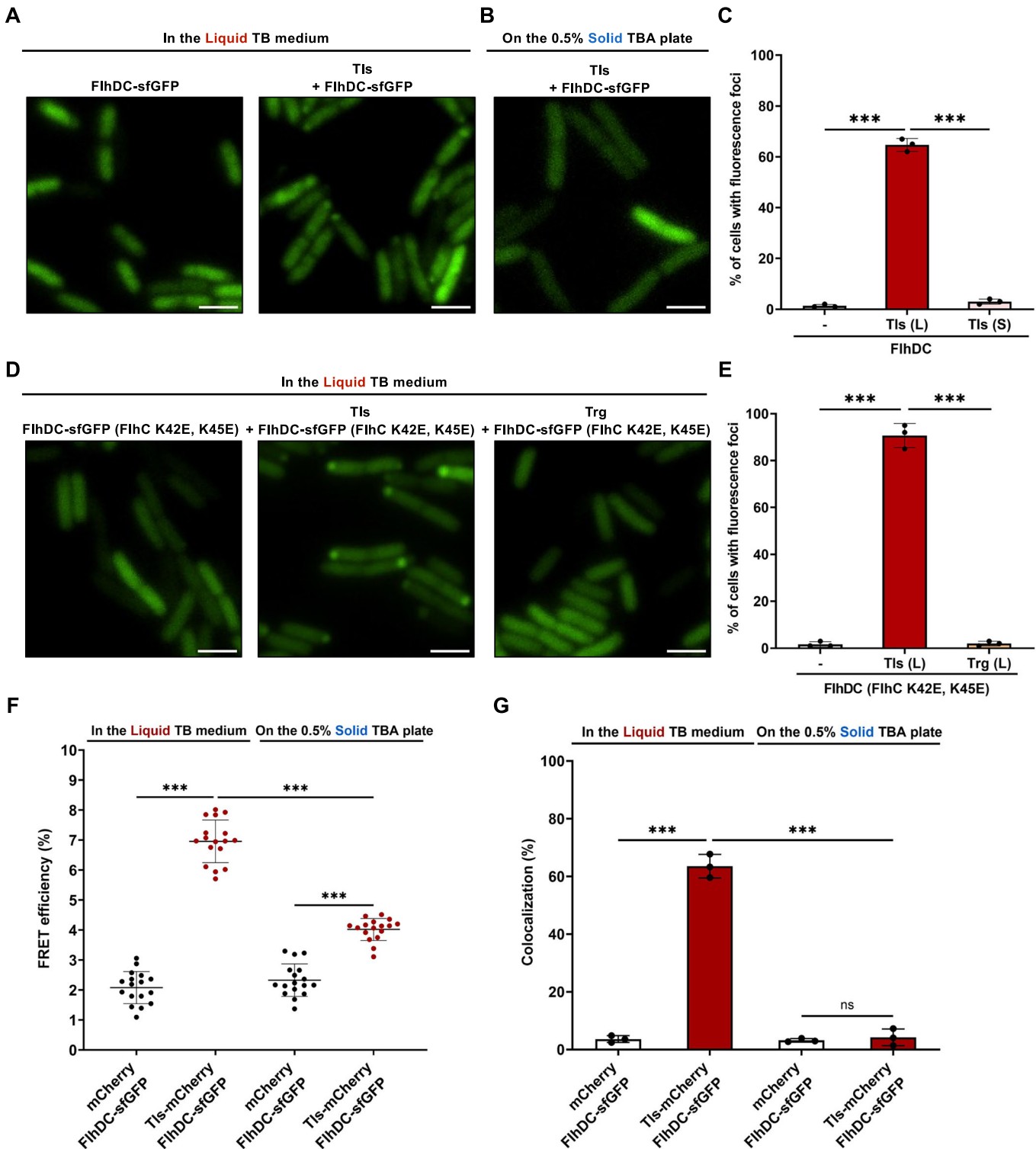

(Fig. EV4C). Consistent with the difference observed in the MG1655 background, there was also an increased number of Tls-mCherry foci and the FlhDC-sfGFP colocalization to these foci in the liquid-grown compared to the surface-grown cultures (Fig. EV4D,E), although the overall localization of Tls-mCherry was less pronounced in the S13 background. Since FRET

experiments in the S13 background had to be performed at the single-cell level (see Methods), we could further compare the data between cells where the Tls-mCherry/FlhDC-sfGFP foci were visible and those where they were not (Fig. EV4C). Although FRET signal was generally stronger in cells showing foci, it was also significantly positive in liquid-grown (but not in surface-grown)

**Figure 6.   Sequestration of FlhDC by Tls.**

(A, B) Localization of the C-terminal sfGFP fusion to FlhC, co-expressed with FlhD as a part of the *flhD flhC-sfgfp* operon (hence called FlhDC-sfGFP) from pBAD33 vector induced with 0.002% L-arabinose, in MG1655 cells grown in liquid TB medium (A) or on 0.5% TBA plate (B). In addition to the FlhDC-sfGFP expression plasmid, cells carried either the empty pTrc99A vector or pTrc99A-Tls induced with 15 μM IPTG, as indicated. Representative images are shown from three biological replicates. Scale bars, 3 μm. (C) Quantification of the number of cells showing fluorescence foci formed by FlhDC-sfGFP in microscopy images, as in (A, B). (D) Localization of FlhDC-sfGFP variant lacking DNA-binding activity, expressed as in (A) in MG1655 cells grown in liquid TB medium, in the absence or the presence of Tls or of Trg expression induced with 15 μM IPTG. Representative images are shown from three biological replicates. Scale bars, 3 μm. (E) Quantification of the number of cells showing fluorescence foci formed in microscopy images as in (D). For quantification of each construct in (C, E), foci were counted in 120 cells from 3 different images, originating from three biological replicates. L, cells grown in the liquid TB medium; S, cells grown on 0.5% TBA plate. Values represent the means and standard deviations. Statistical significance was determined using unpaired two-tailed Student's $t$ test. The $P$ values are denoted as ns ($P > 0.05$), * ($P < 0.05$), ** ($P < 0.005$), *** ($P < 0.001$). ($P$ values from left to right: ***$P = 1.71$E-06, ***$P = 2.40$E-06 (C); $P$ values from left to right: ***$P = 3.01$E-06, ***$P = 3.00$E-06 (E)). (F) FRET measurements of complex formation between FlhDC-sfGFP and Tls-mCherry, co-expressed in MG1655 cells grown either in liquid TB medium or on 0.5% TBA plate. Co-expressed mCherry was used as a negative control. To ensure similar levels of fluorescent protein expression, L-arabinose for FlhDC-sfGFP induction was added to a final concentration of 0.002% to liquid TB and 0.004% to 0.5% TBA plates. Tls-mCherry expression was, respectively, induced with 15 μM or 30 μM IPTG and mCherry expression was, respectively, induced with 30 μM or 200 μM IPTG. FRET efficiency was determined by acceptor photobleaching for the entire cell population in the bleached area as described in "Methods". Symbols represent individual measurements performed for two biological replicates (8–9 measurements per replicate), with the means and standard deviations shown. (G) Quantification of the number of cells showing colocalization of indicated constructs in microscopy images (shown in Fig. EV4A,B) of FlhDC-sfGFP and either mCherry or Tls-mCherry in the cultures used for FRET experiments. For each construct, colocalized foci were counted in 120 cells from three different images, originating from three biological replicates. Values represent the means and standard deviations. Statistical significance was determined using unpaired two-tailed Student's $t$ test. The $P$ values are denoted as ns ($P > 0.05$), * ($P < 0.05$), ** ($P < 0.005$), *** ($P < 0.001$). ($P$ values from left to right: ***$P = 3.10$E-21, ***$P = 3.80$E-16, ***$P = 4.70$E-12 (F); $P$ values from left to right: ***$P = 1.70$E-05, ***$P = 3.40$E-05, ns $= 0.57$ (G). Source data are available online for this figure.

cells where no foci were visible. This result supports our hypothesis that the condition-dependent sequestration of FlhDC-sfGFP by Tls-mCherry does not necessarily require the formation of visible foci and could also occur in smaller complexes in the cytoplasm.

## Tls promotes *E. coli* adherence and invasion in vitro but reduces gut colonization

Because Tls is encoded by members of the B2 phylogroup that are ExPEC pathotypes, we tested the hypothesis that Tls plays a role in how ExPEC strains interact with their host. The most prominent ExPEC pathotype is uropathogenic *E. coli* (UPEC), which is considered to be a highly successful pathogen due to its ability to form chronic reservoirs in the host gut from which it emerges to cause recurrent urinary tract infection (UTI) (Forde et al, 2019; Timm et al, 2024). We thus tested how *tls* deletion would alter the ability of UPEC to establish acute infection in the bladder and from there migrate to the gut and form reservoirs there. For these studies, we used the cystitis isolate UTI89 (Mulvey et al, 2001), a well-established uropathogenic strain which is a close relative of S13 and has functional Tls (Figs. 1C and 2G). When tested in vitro, UTI89 Δ*tls* mutant was significantly less adherent and invasive to urothelial cells compared to the wild-type strain (Fig. 7A,B). Despite the decreased adherence and invasion observed in vitro, subsequent transurethral inoculation of C3H/HeN mice (Appendix Fig. S11A) did not reveal significant differences between *E. coli* colony-forming units (CFUs) in bladder, kidney, or vagina of mice infected with wild-type UTI89 or UTI89 Δ*tls* (Appendix Fig. S11B). Moreover, urine collected from infected animals at 24 h post inoculation indicates no differences in the strain titers, indicating that deletion of *tls* does not influence the ability of UTI89 to proliferate in the bladder (Fig. 7C).

Following the establishment of UTI, UPEC can be found in the vagina and gut of infected humans, where it forms asymptomatic reservoirs that precipitate recurrence (Brannon et al, 2020; Forde et al, 2019). Indeed, analysis of fecal titers revealed the presence of both, UTI89 and UTI89 Δ*tls* in the gut of infected mice, but with a significantly increased number of CFUs for the Δ*tls* (Fig. 7D). Given that the levels of bacteria in the urinary tract were similar for the two strains, these results indicate that the loss of Tls increases migration and/or proliferation of UPEC in the gut.

In order to further investigate whether Tls plays a role in ExPEC proliferation in the gut, we tested the S13 isolate, where the impact of *tls* deletion on flagellar gene expression and motility was even more prominent, in an oral gavage model. In this case, we pre-treated C57BL/6 J specific pathogen-free (SPF) mice with ampicillin 24 h prior to infection to overcome colonization resistance. The mice were then infected with a 1:1 mixture of Δ*tls* and wild-type *E. coli* S13 via oral gavage (Fig. EV5A). Compared to wild-type S13, the S13 Δ*tls* mutant showed a fitness advantage at 48 h post infection (h.p.i.) (Fig. 7E). It was also dominating the cecal content of the mouse gut at 72 h.p.i., as well as distal small intestine (ileum) and proximal colon (Fig. EV5B). Combined, these results suggest the importance of the Tls-mediated control of motility in at least two different host niches (bladder and gut) that are relevant to ExPEC strains.

## Discussion

Motility and chemotaxis can be crucial for bacterial survival and proliferation in their natural environments (Chaban et al, 2015; Colin et al, 2021; Erhardt, 2016; Keegstra et al, 2022; Matilla and Krell, 2018), including colonization of the animal gut (Laganenka et al, 2023). Bacteria encode a diverse, species-specific set of sensory chemoreceptors that differ in their ligand-binding domains but normally produce a highly conserved output, controlling the activity of the chemotaxis pathway (Day et al, 2016; Gavira et al, 2020; Ortega et al, 2017). Here, we describe a novel type of regulation by the receptor-like protein Tls, which apparently evolved from the *E. coli* chemoreceptor Trg to function as a negative regulator of flagellar gene expression in the B2 phylogroup of *E. coli*. Interestingly, regulation by Tls does not require other components of the chemotaxis pathway of *E. coli*, which is in

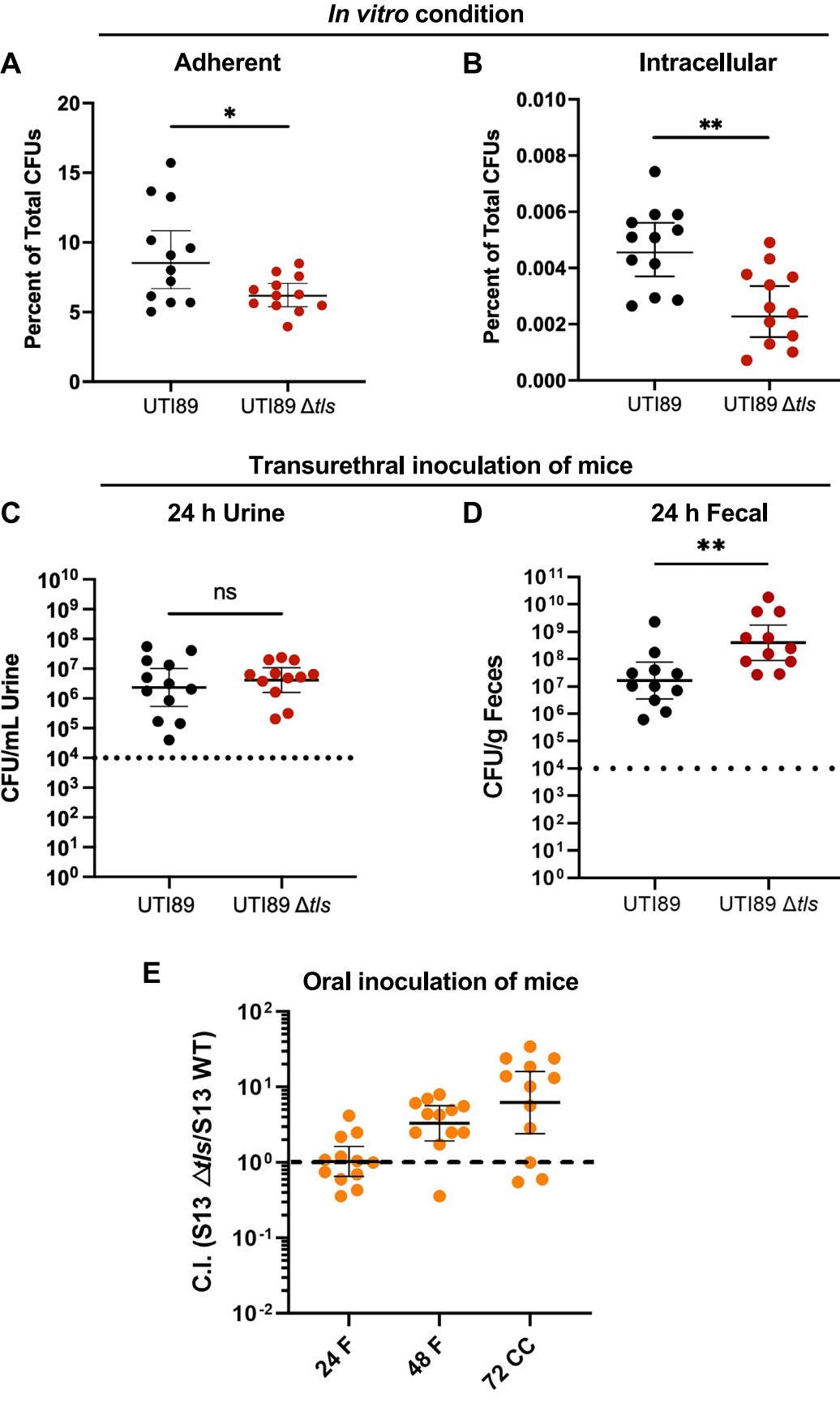

**Figure 7. Importance of Tls in pathogenesis and colonization.**

(A, B) Graphs depict percent adherence (A) and invasion (B) for wild-type UTI89 (black), or the isogenic Δ*tls* mutant (red) using the HTB-9 urothelial tissue culture cell line. Experiments were performed in three biological replicates with four technical replicates in each run. Shown is the percentage of colony-forming units (CFU) formed by adhered or intracellular bacteria from the total bacterial CFU count. The line depicts the geometric mean $+/-$ 95% CI. (C, D) Graph depicts urinalysis (C) and fecal sample (D) titers of UTI89 (black) and the isogenic Δ*tls* mutant (red) for each mouse collected at 24 h post infection. Urine was collected from each mouse and plated as previously described. Urinalysis is used to track shedding of bacteria from the lumen over time and serves as an indicator for dissemination to the gut. Fecal samples were collected from each mouse to track UPEC proliferation in the gut. To facilitate urinalysis and fecal collection, each mouse was tagged with an identifying ear notch. The same 11–12 mice were tracked in the experiments shown in (C, D). Dashed line indicates the clinical threshold for UTI in humans. The line depicts the geometric mean $+/-$ 95% CI. Statistical significance was determined using unpaired two-tailed Mann–Whitney *U* test. The *P* values are denoted as ns ($P > 0.05$), * ($P < 0.05$), ** ($P < 0.005$), *** ($P < 0.001$). (*$P = 0.0372$ (A); **$P = 0.0014$ (B); ns $= 0.4428$ (C); **$P = 0.0049$ (D)). (E) Competitive index (C.I.) of S13 Δ*tls* compared to the wild-type S13 strain, determined as the ratio of CFU counts, in the fecal samples (denoted as F) collected at 24 h or 48 h post gavage infection (h.p.i.), and in the cecal content (denoted as CC) at 72 h.p.i. Analyses were performed using 12 mice (individual dots). The line depicts the geometric mean $+/-$ 95% CI, and the dashed line indicates the CI value of 1. Source data are available online for this figure.

contrast to a few previously described cases when the entire pathway was evolutionary reprogrammed to perform a new function (Huang et al, 2019; He and Bauer, 2014).

Our results suggest that Tls inhibits flagellar gene expression by sequestering FlhDC away from the chromosome, which apparently involves complex formation between the two proteins. These complexes at least partly localize to the discrete foci formed by Tls that, similar to other supramolecular protein assemblies in bacteria (Hoang et al, 2024; Winkler et al, 2010), exhibit polar localization. However, the sequestration of FlhDC can also include smaller cytoplasmic complexes that prevent it from DNA binding, and complex formation between Tls and FlhDC was indeed observed even in cells that showed no visible foci. This represents a novel mechanism to modulate FlhDC activity at a post-translational level, although such sequestration of transcription factors has been previously described in other regulatory systems in bacteria and eukaryotes (Gibson et al, 2022; Hoppe et al, 2001; Lee et al, 2000; Vik and Rine, 2000).

We further observed that the Tls-mediated inhibition of flagellar gene expression, including sequestration of FlhDC, as well as localization of Tls as discrete foci, only occurs during growth in liquid medium but not in the soft or on the semi-solid agar. This suggests that the Tls-mediated sequestration of FlhDC depends on the properties, possibly mechanical, of the cell growth environment, and that it is either mediated or modulated by other cellular factors that remain to be identified. Indeed, mechanosensitive repression of flagellar gene expression during growth in liquid, but not in the porous medium, was recently described for several *E. coli* strains that belong to the B2 phylogroup, including S13 (Laganenka et al, 2020). The underlying molecular mechanism remained unclear, although RflP was shown to be required to reduce the basal level of flagellar gene expression to enable mechanosensing (Laganenka et al, 2020). Our results suggest that such mechanosensitive regulation mediated by Tls acts in addition to, but independently of, RflP to inhibit FlhDC activity. Although the exact mechanism of mechanosensing by Tls remains to be elucidated, it apparently does not require flagella or the chemotaxis system, as the Tls-mediated repression and its surface-dependent (de)localization are maintained in the *fliA* and *flhC* mutant strain backgrounds. It might instead rely on one of the other known mechanisms of mechano- or surface-sensing (Gordon and Wang, 2019).

Interestingly, the short N-terminal fragment of Tls (TlsN) can at least partly mediate this function not only when fused to the cytoplasmic fragment of Trg but also to other proteins. Our results indicate that this functionality may require TlsN fusion proteins to be at least dimeric. Such dimerization might be normally mediated by the Trg-derived fragment of Tls, but the cytoplasmic domains of other chemoreceptors or other dimeric proteins could also fulfill this function. How Tls forms higher-order complexes and recruits FlhDC remains to be elucidated, but the observed high dynamics of protein exchange within the observed Tls complexes is characteristic of biomolecular condensates. Given the importance of a positively charged arginine (or lysine) for the structure formation and function of TlsN, we could hypothesize that nonspecific electrostatic interactions with nucleic acids, especially RNA, as known for other biomolecular condensates (Boeynaems et al, 2018; Wang et al, 2018), might contribute to the complex formation.

Tls is only present in the B2 phylogroup of *E. coli* that includes major UPEC and other ExPEC strains. While flagella and motility have been shown to be partially contributing to the movement of UPEC from the bladder lumen to the kidney (Lane et al, 2007; Wright et al, 2005), no studies have evaluated how the population may be controlling the production of flagella during infection. In addition, no studies—until this one—have evaluated how changing motility influences pathogen reservoirs. Finally, it is important to note that during acute infection, intra-strain heterogeneity arises in UPEC as a function of unique microenvironments, including the intracellular space (Beebout et al, 2019; Brannon et al, 2024). Our data suggest that the conditional regulation by Tls might coordinate the dual function of motility in this versatile lifestyle. Adherence and invasion potentiate UPEC proliferation inside the bladder cell, and under these conditions, the suppression of motility might be beneficial, explaining the lower adherence of the *tls* strain. However, bacterial cells may benefit from motility to enable dissemination from the bladder lumen to the kidney or the gut. Motility in the gut further enables *E. coli* to use chemotaxis to accumulate toward nutrient-abundant niches (Colin et al, 2021; Laganenka et al, 2023). These potential benefits of motility can help to explain why flagellar gene expression becomes derepressed in the porous environment, such as that of the gut, and why the deletion of *tls* provides a competitive fitness advantage for the proliferation of *E. coli* in the murine gut. For UPEC, this ability to form chronic reservoirs in the host gut from which it emerges to cause recurrent infection may make a major contribution to its success as a pathogen (Forde et al, 2019; Timm et al, 2024) that accounts worldwide for ~405 million urinary tract infections on an annual basis (Deng et al, 2024). Given that Tls appears to control

adherence in the bladder and migration or proliferation in the gut, it is possible that UPECs (and other ExPECs) benefit from Tls to "toggle" between infection and reservoir niches. Thus, the observed regulation by Tls might be an important contributor to the ExPEC pathogenicity.

# Methods

### Reagents and tools table

| Reagent/resource | Reference or source | Identifier or catalog number |
|---|---|---|
| **Experimental models** | | |
| *E. coli* S13 | Leibniz Institute DSMZ - German Collection of Microorganisms and Cell Cultures (Braunschweig, Germany) | DSM 10719 |
| *E. coli* S13 Δ*tls* (Kanamycin sensitive) | This work | |
| *E. coli* S13 Δ*rflP* (Kanamycin sensitive) | Laganenka et al, 2020 | |
| *E. coli* S13 Δ*flhC* | This work | |
| *E. coli* UTI89 | Mulvey et al, 2001 | |
| *E. coli* UTI89 Δ*tls* (Kanamycin sensitive) | This work | |
| *E. coli* MG1655 | Blattner et al, 1997 | |
| *E. coli* MG1655 Δ*fliA* | This work | |
| *E. coli* MG1655 Δ*flhC* | Rudenko et al, 2019 | |
| *E. coli* RP437 | Parkinson and Houts, 1982 | |
| *E. coli* RP437 Δ*tar tsr trg tap aer* | Ames et al, 2002 | |
| *E. coli* RP437 Δ*cheR* | Lovdok et al, 2007 | |
| Mouse line: C57BL/6 | Jackson Laboratory | JAX:00066 |
| Mouse line: C3H/HeN | Inotiv (formerly Envigo) | 040 |
| **Recombinant DNA** | | |
| pKD46 | Datsenko and Wanner, 2000 | |
| pCP20 | Cherepanov and Wackernagel, 1995 | |
| pUA66 | Zaslaver et al, 2006 | |
| pTrc99A | Amann et al, 1988 | |
| pBAD33RBS | Yuan et al, 2017 | |
| pVS1962 | Ni et al, 2017 | |
| pVS1963 | Zaslaver et al, 2006 | |
| pVS1970 | Ni et al, 2017 | |
| pTrc99A-Tls | This work | |

| Reagent/resource | Reference or source | Identifier or catalog number |
|---|---|---|
| pTrc99A-TlsN-TrgΔN (TrgΔN, aa 400-546 of Trg) | This work | |
| pTrc99A-TlsN-TarΔN (TarΔN, aa 390-553 of Tar) | This work | |
| pTrc99A-TlsN-TsrΔN (TsrΔN, aa 392-551 of Tsr) | This work | |
| pTrc99A-TlsN-TapΔN (TapΔN, aa 388-533 of Tap) | This work | |
| pTrc99A-TlsN-AerΔN (AerΔN, aa 382-506 of Aer) | This work | |
| pTrc99A-TlsN | This work | |
| pTrc99A-Tls-sfGFP | This work | |
| pTrc99A-TlsN-sfGFP | This work | |
| pTrc99A-TlsN-sfGFP(N149Y) | This work | |
| pTrc99A-TlsN-sfGFP(R206A) | This work | |
| pVS2890 | Scarinci and Sourjik, 2023 | |
| pTrc99A-TlsN-mCherry | This work | |
| pTrc99A-M(Gly)$_8$-mCherry | This work | |
| pTrc99A-TlsN (R4D)-mCherry | This work | |
| pTrc99A-TlsN (R4K)-mCherry | This work | |
| pTrc99A-TlsN*-mCherry | This work | |
| pTrc99A-Tls-mCherry | This work | |
| pTrc99A-mCherry-Tls | This work | |
| pTrc99A-Trg | This work | |
| pBAD33RBS-FlhDC-sfGFP | This work | |
| pBAD33RBS-FlhDC-sfGFP (FlhC K42E, K45E) | This work | |
| pM965 | Stecher et al, 2004 | |
| **Antibodies** | | |
| Anti-GFP antibody produced in a rabbit | Sigma-Aldrich | SAB4701015 |
| Goat Anti-Rabbit IgG Antibody, HRP-conjugate | Sigma-Aldrich | 12-348 |
| **Oligonucleotides and other sequence-based reagents** | | |
| Cloning primer sequences | See Appendix Table S1 | |
| **Chemicals, enzymes, and other reagents** | | |
| Bacto Tryptone | Gibco | 211705 |
| Sodium chloride | Carl Roth | 3957.1 |
| Yeast extract | Gibco | 212720 |
| Bacto Agar | BD | 214010 |

| Reagent/resource | Reference or source | Identifier or catalog number |
|---|---|---|
| Agarose HEEO ultra-quality | Carl Roth | 2269.1 |
| Kanamycin | Sigma-Aldrich | 25389-94-0 |
| Ampicillin | Carl Roth | 29T3.1 |
| Chloramphenicol | Sigma-Aldrich | 56-75-7 |
| IPTG | Carl Roth | 2316.1 |
| Poly-dimethylsiloxane (PDMS) | Dow Corning Sylgard184 | 101697 |
| SacI-HF restriction enzyme | New England Biolabs | R3156S |
| HindIII-HF restriction enzyme | New England Biolabs | R3104S |
| Gibson Assembly Master Mix | New England Biolabs | E2611L |
| Q5 Site-Directed Mutagenesis Kit | New England Biolabs | E0554S |
| Isoflurane | McKesson | 803250 |
| MacConkey Agar | Fisher Scientific | B12306 |
| **Software** | | |
| Geneious Prime | Geneious Prime Software | https://www.geneious.com/ |
| ImageJ | National Institutes of Health | https://imagej.nih.gov/ij/ |
| NIS Elements AR software | Nikon | https://www.microscope.healthcare.nikon.com/products/software/nis-elements/nis-elements-advanced-research |
| R software | R Foundation Statutes | https://www.r-project.org/ |
| GraphPad Prism 10 | Graphpad Software | https://www.graphpad.com/ |
| **Other** | | |
| Infinite M1000 PRO plate reader | TECAN | |
| TI Eclipse phase-contrast microscope | Nikon | |
| Ti-E inverted fluorescence microscope | Nikon | |
| Chemiluminescence imager | Intas | |
| LSM 880 confocal laser scanning microscope | Carl Zeiss | |
| Stainless steel beads | Qiagen | 69989 |
| Tissue lyzer | Omni | TH115 |

## Methods and protocols

### Bacterial strains, plasmids, and growth conditions

The strains and plasmids used in this study are listed in the Reagents and Tools Table. Unless specified otherwise, cultures of *E. coli* strains were grown at 37 °C in liquid tryptone broth (TB) medium (10 g tryptone and 5 g NaCl per liter) or in lysogeny broth (LB) medium (10 g tryptone, 10 g NaCl, and 5 g yeast extract per liter), supplemented with antibiotics (100 µg/mL ampicillin and 50 µg/mL kanamycin) where necessary and indicated

concentrations of inducers, isopropyl β-D-1-thiogalactopyranoside (IPTG) and/or L-arabinose. Gene deletions were obtained via PCR-based inactivation (Datsenko and Wanner, 2000), and Km^r cassettes were eliminated via FLP recombination (Cherepanov and Wackernagel, 1995). Expression plasmids were constructed using Gibson assembly reaction in NEBuilder HiFi DNA Assembly Master Mix (New England BioLabs). Point mutations were introduced using Q5® Site-Directed Mutagenesis Kit (New England BioLabs). Primers used for cloning and mutagenesis are listed in Appendix Table S1.

### Sequence analysis

For the completely sequenced *E. coli* strains, nucleotide and amino acid sequences of *trg* region were extracted from the NCBI (Sayers et al, 2022) and Microbial Signal Transduction Database (MiST4.0) (Gumerov et al, 2024). For the strains from the ECOR collection (Ochman and Selander, 1984; Patel et al, 2018), sequences of *trg* region were identified using BLAST (Altschul et al, 1990) in the whole-genome shotgun contigs database limited by WGS Project with default threshold values. The nucleotide sequences of *trg* region from total 115 *E. coli* stains were aligned using Clustal Omega (Sievers et al, 2011), then the resulting tree was imported into iTOL (Letunic and Bork, 2021) to generate an unrooted phylogenetic tree.

### Bacterial motility characterization

Day cultures of *E. coli* strains were inoculated by diluting overnight cultures 1:100 in 10 mL fresh TB and grown at 37 °C in a rotary shaker at 200 r.p.m. for 3 h 15 min, with the optical density at 600 nm ($OD_{600}$) reaching 0.5–0.8 depending on the strain. Where indicated, expression of Tls was induced with 15 or 50 µM IPTG. The cultures were subsequently diluted 1:10 in motility buffer (6.15 mM $K_2HPO_4$, 3.85 mM $KH_2PO_4$, 0.1 mM EDTA, 67 mM NaCl, pH 7.0) and a 2 µl drop was sandwiched between two glass slides separated by a 140-µm thick spacer, with additional grease sealing to prevent drop evaporation. A first movie was recorded within 10 min of making the sample, to minimize effects of sedimentation, in the center of the drop away from surfaces, at ×10 magnification using a phase-contrast microscope (Nikon TI Eclipse) equipped with a complementary metal-oxide semiconductor (CMOS) camera EoSens 4CXP (exposure time, 1 ms; 2×2-binned pixels of 1.4 µm) at a rate of 100 frames per second (fps) for 100 s. Several hundred cells were imaged at any given time, with the exact number of cells in the field of view varying over time, as cells were swimming in and out. These movies were analyzed by Differential Dynamic Microscopy (DDM) (Wilson et al, 2011) as described previously (Lisevich et al, 2025) to extract the fraction of swimming cells and their swimming speed. To measure the tumbling rate of swimmers, two movies were then recorded after sedimentation took place, halfway between the two glass surfaces and close to the top glass surface, at 50 fps for 40 s with the same camera (with non-binned pixels of 0.7 µm) and analyzed using our custom Particle_Tracking_2 ImageJ Plugin (described in detail at https://gitlab.gwdg.de/remy.colin/particletracking2). In short, cells were identified in each frame via intensity thresholding of background-subtracted images and linked into trajectories via identification of closest neighbor in the next frame. Trajectories ($r_i(t)$) longer than a second (50 frames) were categorized as swimmer or non-swimmer via a threshold on the radius of gyration

of the trajectory $r_{g,i} = \langle (r_i(t) - \langle r_i(t) \rangle_t)^2 \rangle_t / T_i$, with $T_i$ the trajectory duration (Appendix Fig. S3B, $r_{g,i} > 0.2$ for swimmers). Swimmer trajectories were split between runs and tumbles via a criterion on motion direction persistence (using the default software settings). The tumbling bias is the swimmer-population-averaged fraction of time spent tumbling. Swimming speed was also measured and was consistent with DDM results. The images of Fig. 2B were produced by recording a movie at 20 fps close to the bottom surface just after making the sample. Images were then normalized by dividing each frame by an image of the background illumination under the same conditions, and the greyscale was inverted in order for the cells to appear bright. A projection of the temporal maximum of each pixel was computed to produce an image of the cell trajectories. Measurement of chemotactic response to α-methyl-D,L-aspartate (MeAsp) was performed as described previously (Colin et al, 2019; Colin et al, 2014). Briefly, 1 mL of the day culture was washed trice in motility buffer by gentle centrifugation (4000 rpm for 5 min), supernatant removal and resuspension, and finally supplemented with 1% glucose and 0.01% Tween-80. A gradient-generating device, made of two large reservoirs connected by a channel with 2 mm length, 1 mm width and 70 μm height, was produced by bounding molded poly-dimethylsiloxane (PDMS, Dow Corning Sylgard184) pealed from an SU8-based mold to a microscopy glass slide after oxygen plasma treatment. The device was prefilled with deionized sterile water 20 min after bounding to retain surface hydrophilic properties. Washed cells supplemented with either 100 μM or no MeAsp were loaded in the respective reservoirs, the sample was sealed with grease, and a linear gradient of MeAsp was allowed to form in the channel for 2 h. Cell motion was recorded as for the DDM measurements, in the center of the channel at mid-height. Movies were analyzed with DDM to compute the fraction of swimmer (Phi) and the swimming speed ($V_0$) of swimmers, and with phase differential microscopy (PhiDM) to compute the population-averaged drift velocity ($V_d$) towards the 100 μM MeAsp reservoir (Colin et al, 2019; Colin et al, 2014). The average chemotactic drift velocity of the swimmers was estimated as $V_{ch} = V_d/Phi$, since the non-motile cells have zero drift, and the chemotactic bias of the swimmers as bias = $V_{ch}/V_0$.

### Soft-agar motility assay

To test the spreading in the porous environment, spreading in soft (0.27%) agar containing TB was used. Two microliters of the bacterial day cultures grown to $OD_{600} = 0.6$ as described above were spotted onto the surface of the soft agar plate containing respective antibiotics and inducers. Plates were incubated at 37 °C for approximately 6 h, photographed using Nikon D5200 camera, and evaluated using ImageJ software.

### Promoter reporter assay

Promoter activities of flagellar genes (*flhD*, *fliA*, and *fliC*) were assayed using plasmid-based reporters containing the respective promoter regions fused to *gfp* (Laganenka et al, 2020). Bacterial overnight cultures were inoculated into fresh TB containing indicated concentrations of IPTG at an initial $OD_{600}$ of 0.05 and grown in 96-well plates (Greiner Bio-One) in a plate reader (Infinite M1000 PRO, TECAN) at 37 °C with shaking (200 r.p.m.). $OD_{600}$ and GFP fluorescence were measured every 10 min. Reporter activity in the log phase of growth was calculated as GFP fluorescence normalized by the value of

$OD_{600}$. For the bacterial cultures grown on the semi-solid medium (0.5% agar TB plates), cells were grown for 3.5 h, washed from the plate using 2 ml TB medium, and fluorescence was measured in the plate reader and normalized by the value of $OD_{600}$ as described above.

### Whole proteome analysis

For the proteome analysis, *E. coli* cultures were grown in TB medium as above until $OD_{600}$ of 0.6. Cells were then harvested by centrifugation (4000 r.p.m., 5 min) and washed twice with 5 ml cold PBS, and the pellets were stored at −20 °C. For the MG1655 strain, 15 μM IPTG was added during culture growth to induce expression of Tls. The collected samples were subsequently subjected to proteomic analysis using liquid chromatography-mass spectrometry (LC-MS). The instrumental settings have been described in detail previously for the applied data-independent (DIA; Perez-Lopez et al, 2025; Scarinci et al, 2024) and data-dependent acquisition mode (DDA; Wimmi et al, 2024). In short, peptide samples were analyzed on an Ultimate 3000 RSLC nano system connected to an Exploris 480 via a nanospray Flex ion source (all Thermo Fisher Scientific). Peptides separation was carried out at 300 nL/min with an acetonitrile gradient increasing from 6-35% acetonitrile over 60 and 90 min, respectively. Due to new workflow implementation, parts of the experiments were acquired in DDA and DIA mode on the Exploris-MS system.

For DDA analysis, the following parameters were used: MS 1 scan at a resolution of 60,000 full width at half maximum (at $m/z$ 200) and MS/MS scan acquisition of the most intense ions within 1 s (cycle 1 s) with exclusion of singly charged ions. The ion accumulation time was set to 50 ms (MS) and 50 ms at 17,500 resolution (MS/MS). The automatic gain control (AGC) was set to $3 \times 10^6$ for MS1 scan and $2 \times 10^5$ for MS/MS scans. The quadrupole isolation was 1.5 $m/z$, collision was induced with an HCD collision energy of 27%. DDA MS raw data was then analyzed with MaxQuant (Cox and Mann, 2008) executed in standard settings. The search criteria were set as follows: full tryptic specificity was required (cleavage after lysine or arginine residues); two missed cleavages were allowed; carbamido-methylation (C) was set as fixed modification; oxidation (M), deamidation (N,Q).

For DIA analysis, full MS 1 was set to 120,000 (at $m/z$ 200). AGC target value for fragment spectra was set at 3000%. 45 windows of 14 Da were used with an overlap of 1 Da between $m/z$ 320–950. Resolution was set to 15,000 and fill time to 22 ms. Stepped HCD collision energy of 25, 27.5, 30% was used. MS1 data were acquired in profile, MS2 DIA data in centroid mode. Analysis of DIA data was performed using the DIA-NN version 1.8 (Demichev et al, 2020).

Although no reference database for *E. coli* S13 is available, using the UniProt protein database of a closely related *E. coli* UTI89 resulted in nearly full coverage of the analyzed proteome.

### Fluorescence microscopy

*E. coli* cultures were grown as described above for motility experiments, harvested by centrifugation (4000 r.p.m., 5 min), washed, and resuspended in a tethering buffer (6.15 mM $K_2HPO_4$, 3.85 mM $KH_2PO_4$, 0.1 mM EDTA, 1 μM L-methionine, and 10 mM sodium lactate, pH 7.0) and applied to a thin agarose pad (1% agarose in tethering buffer). For bacterial cultures grown on

semi-solid medium (0.27% or 0.5% agar TB plates), cells were collected from the plate by washing it with 2 ml tethering buffer before applying to the agarose pad. Bacterial cells grown in soft (0.27%) agar were scraped gently with an inoculation loop and resuspended in 15 μL tethering buffer. The images were acquired using a Nikon Ti-E inverted fluorescence microscope with a x100 objective lens and a Lumencor SOLA-SEII equipped with an Andor Zyla sCMOS camera. All images shown in one figure were acquired under identical conditions and displayed using the same settings. Images were subsequently evaluated using ImageJ software (Wayne Rasband, National Institutes of Health, USA).

### Western blot analysis

Cell lysates were prepared as described before with modifications (Yuan et al, 2017). Specifically, E. coli cultures were grown at 37 °C to $OD_{600}$ of 0.6 in TB liquid medium on a rotary shaker or on 0.5% TB agar plates, supplemented with the antibiotic (100 μg/mL ampicillin) and inducer (15 μM IPTG). Samples were collected by pelleting equal amounts of cells, and the pellets were stored at –20 °C. For the immunoblot analysis, the pellets were resuspended in SDS-loading buffer and lysed by boiling for 10 min at 95 °C. After equal amounts of lysates were loaded on a 10% SDS polyacrylamide gel, proteins were separated by gel electrophoresis and then transferred to a nitrocellulose membrane. Tls-sfGFP fusion protein was detected with specific anti-GFP primary antibody (Sigma) and horseradish peroxidase-conjugated secondary antibody (Sigma). Protein bands were detected by chemiluminescence using an ADVANCED imager (Intas). Total protein stain of the PVDF membrane using Revert 700 Total Protein Stain solution (LI-COR) was used as a loading control.

### Fluorescence recovery after photobleaching (FRAP) assay

E. coli cultures were grown as described above in TB supplemented with 15 μM IPTG at 37 °C, harvested at $OD_{600}$ of 0.6 by centrifugation (4000 r.p.m. for 5 min), washed with tethering buffer, and applied to agarose pads. FRAP measurements were performed on a LSM 880 confocal laser scanning microscope (Carl Zeiss Microscopy) equipped with a C-Apochromat 40×/1.2 water immersion objective. Fluorescence was excited at 488 nm (Argon laser, 25 mW, nominal power <0.1%) and emission was collected in the 490–580 nm range. Images were routinely acquired using ×20 zoom, 100 × 100 pixels (pixel size: 0.1 μm). After the acquisition of 10 pre-bleaching frames, photobleaching was induced by a 75–100% laser pulse in a 12 × 12 pixels area. For data analysis, the bleaching spot, the target cell, and a reference area for background subtraction were defined as ImageJ regions of interest (ROIs). The fluorescence intensity of the bleached area was normalized by the value of prebleach intensity and by the integral fluorescence of the entire cell in the same image to correct for additional photobleaching during image acquisition.

### FRET measurements using acceptor photobleaching microscopy

E. coli cultures were grown as described above for motility and microscopy measurements, with indicated levels of induction by L-arabinose and IPTG. One milliliter culture was washed and resuspended in 10 μL tethering buffer. For cells growing on surfaces, cells were washed away from the surface using 1.5 mL tethering buffer, and washed and resuspended in 120 μL tethering buffer. Three microliters of the cells were then applied on the

coverslip and covered by an agarose pad. Measurements of the FRET efficiency by acceptor photobleaching were performed as described previously (Yadavalli et al, 2020) with slight modifications. Briefly, acceptor photobleaching FRET was performed using a dual-layer Nikon Ti-E inverted fluorescence microscope equipped with a fluorescence lamp (X-cite Exacte; Lumen Dynamics), a perfect focus system (PFS), and NIS Elements AR software (version 4.40 Nikon). Images were acquired through a 40× Plan Apo 0.95-numerical-aperture (NA) objective in the sfGFP (donor) and mCherry (acceptor) channels; excitation power was adjusted both by controlling the fluorescence lamp output and with neutral-density (ND) filters. Specifically, the donor was excited at 482/18 nm and its emission was detected at 525/50 nm, and the acceptor was excited at 585/29 nm, while its emission was detected at 647/57 nm. The acceptor photobleaching was conducted using 593.5 nm laser. Images were recorded using an iXon 897-X3 EM-charge-coupled-device (CCD) camera (Andor). For each position, sequences of images of isolated sample areas were acquired with the following protocol: (i) first, 2 images were taken in the acceptor channel (exposure time 1 s); (ii) then, 40 images (exposure time 500 ms) were taken in the donor channel, followed by (iii) 12 s of acceptor photobleaching (no image acquisition); afterward, (iv) 40 images (exposure time 500 ms) were taken in the donor channel, and then (v) 2 images (exposure time 1 s) were taken in the acceptor channel. In order to avoid donor fluorescence recovery from interrupted illumination, the sample was continuously illuminated with a 482/18-nm excitation light, even during step iii. FRET efficiency was calculated as the increase in the donor signal upon acceptor photobleaching divided by the donor signal after acceptor photobleaching, based on the background-subtracted integrated fluorescence intensity of cells in the entire bleached area. In order to correct for the donor photobleaching present during steps ii to iv, we performed linear fitting (RStudio) of the donor fluorescence signal versus time for both pre- and postbleaching curves.

Since the FRET setup described above was not accessible for microscopy of the S13 strain for biosafety reasons, FRET measurements in that background were performed at the single-cell level, as described previously with slight modifications (Chen et al, 2025). Briefly, samples were prepared as above, except that cells were resuspended at lower density, in 0.8–1 mL tethering buffer, and 4 μL of this suspension was applied on the coverslip and covered by an agarose pad. The measurement was conducted using Visitron TIRF/FRAP microscope (Visitron, Puchheim, Germany) equipped with a 2D FRAP Scanner, in which the Nikon Eclipse Ti-E microscope was equipped with CoolLED pE-4000 light source and operated by VisiView software (3.3.0.6). Images were acquired through Nikon 100× Apo TIRF NA 1.49 oil immersion objective and iXon Ultra-888 EMCCD camera (Andor). The donor was excited at 474/27 nm, and its emission was detected at 525/39 nm, and the acceptor was excited at 575/25 nm while its emission was detected at 645/90 nm. The acceptor photobleaching was conducted pixel by pixel using FRAP Scanner with 561 nm laser. For each position, 25 images were acquired in the donor channel (exposure time 400 ms), followed by 1.8 s of acceptor photobleaching (no image acquisition), and acquisition of another 25 images in the donor channel (exposure time 400 ms). Acceptor images were acquired before and after photobleaching (exposure time 200 ms). The outline of each bleached cell was manually drawn using

ImageJ. Upon background subtraction, the data were analyzed as described before (Chen et al, 2025).

### Bacterial adherence and invasion assays in the tissue culture

For tissue culture experiments, the 5637 (ATCC HTB-9) bladder epithelial cell line was used as we previously described (Brannon et al, 2024). Epithelial cells were grown to at least 90% confluency in 24-well plates. Prior to inoculation with *E. coli*, new RPMI 1640 media supplemented with 10% FBS was added to the cells. Wild-type *E. coli* UTI89 or its Δ*tls* mutant were added to an approximate multiplicity of infection (MOI) of 5, based on HTB-9 cell density, to triplicate wells. Plates were incubated at 5% $CO_2$ and 37 °C for 2 h. For cell lysis, a final concentration of 0.1% Triton X-100 was used. One set of wells was lysed to determine the total number of bacteria within the well. The other two sets were washed with 0.5 mL of PBS three times. Adherent and intracellular total bacterial titers were enumerated in a set of wells that was immediately lysed after the washes. The final set of wells was gently washed with PBS with 100 µg/mL of gentamicin (Life Technologies Co., Grand Island, NY, USA) for 2 h to kill all extracellular/adherent bacteria. Following the 2 h gentamicin incubation, wells were washed two more times with 1 mL of PBS and treated with 0.1% Triton X-100 to lyse the HTB-9 cells and enumerate the intracellular *E. coli*. The percentage of *E. coli* adherence and invasion was calculated as a percentage of the total number of bacteria.

### Murine urinary tract infection model

Animal studies using the murine infection model were conducted at Vanderbilt University Medical Center (VUMC) and were approved by the VUMC Institutional Animal Care and Use Committee (IACUC) (protocol number # M1800101-01). Animals were purchased from Envigo (*E. coli*-associated C3H/HeNHsd) and acclimated at the Vanderbilt University Medical Center (VUMC) facilities for at least 4 days before inoculation. Only female animals were used for this study at 7- to 8-weeks of age. Power analysis software from Statistics Solutions® is used to calculate the required number of animals per condition in each independent experiment. Specifically, we seek to demonstrate a statistically significant difference in CFU counts that corresponds to 75% of the mean between control and test conditions, assuming an S.D. that is 20% of the mean. With a two-sided test and a confidence level of 90%, we determine that $n = 7$ is the smallest number of animals required per condition in each independent experiment. Accordingly, to evaluate acute infection outcomes, 10 mice were infected per-condition-per-experiment to include three mice for microscopy purposes. Prior to the inoculation, all animals were transferred in cages of 5 by veterinary personnel with no connection to the study. The person infecting the animals was blinded to the inoculum identity. To prepare the inoculum, *E. coli* UTI89 and the isogenic Δ*tls* mutant were grown overnight in 5 mL LB broth with shaking, at 37 °C. Cultures were then diluted 1:1000 into 10 mL fresh LB and grown at 37 °C statically for 24 h. Cultures were subsequently diluted 1:1000 into 10 mL fresh LB and propagated at 37 °C statically for another 24 h. Cultures were normalized in sterile PBS to a final $OD_{600} = 3.4$ and diluted tenfold. This dilution results in an inoculum size from this suspension is $10^7$ CFU per 50 µl. For transurethral inoculation, female C3H/HeN mice (7–8 week old) purchased from Envigo were used. Each mouse was transurethrally inoculated under anesthesia, as

previously described (Hung et al, 2009) with 50 µL containing $10^7$ CFUs of bacteria. Mice were euthanized at 48 h post infection for CFU analysis. Bladders, vaginal membranes, and kidneys were removed, suspended in 1 mL sterile PBS, homogenized with an Omni tissue homogenizer, and serially diluted for CFU/organ enumeration. Urine and fecal samples were collected from mice at 24 h post infection.

### Mouse oral infection

Mouse oral infection experiments were conducted at ETH Zurich and were approved by Tierversuchskommission, Kantonales Veterinäramt Zürich under license ZH158/2019. C57BL/6 (Jackson Laboratory, catalog no. JAX:00066) mice were maintained under specific pathogen-free (SPF, free of *Enterobacteriaceae* members) conditions at the EPIC facility at ETH Zurich. The sample size was determined based on previous experience with CI experiments. Normally, 5–8 animals are enough to determine the statistical significance of the obtained data. The 8- to 12-week-old mice of both sexes were randomly assigned to experimental groups (up to 5 mice per cage). To allow for *E. coli* colonization, mice were orally pre-treated with 20 mg ampicillin 24 h prior to infection. *E. coli* S13 and its isogenic Δ*tls* mutant were therefore transformed with the fitness-neutral pM965 (Laganenka et al, 2023; Stecher et al, 2004) prior to infection. Overnight cultures of both strains were grown in TB at 37 °C with shaking, diluted in fresh TB (1:100), and incubated at 37 °C with shaking for 3 h (final $OD_{600} = 0.5–0.6$). The cells were washed in PBS, and the mice were infected with $5 × 10^7$ CFU of *E. coli* WT and Δ*tls* mix (1:1). Fecal samples were collected and suspended in 500 µl PBS at 24 and 48 h post *E. coli* infection, followed by homogenization in a Tissue Lyzer (QIAGEN). The samples were then plated out on MacConkey or LB agar plates with appropriate antibiotics to determine the CFU numbers of the wild-type and the knockout strains. At 72 h.p.i., mice were euthanized by $CO_2$ asphyxiation, and contents of distal small intestine, cecum and proximal colon were harvested, suspended in 500 µl PBS, homogenized and plated as described above. The competitive index (CI) of *E. coli* S13 WT and Δ*tls* was calculated as the ratio between CFU (Δ*tls*) and CFU (WT), divided by the ratio of both strains in the inoculum.

## Data availability

The mass spectrometry proteomics data have been deposited to the ProteomeXchange Consortium via the PRIDE partner repository (Perez-Riverol et al, 2025) (http://www.ebi.ac.uk/pride) with the dataset identifier PXD064677. Code for particle tracking is available at GitLab (https://gitlab.gwdg.de/remy.colin/particletracking2). All other data are within the paper and its supporting information files.

The source data of this paper are collected in the following database record: biostudies:S-SCDT-10_1038-S44318-025-00595-x.

## Peer review information

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

## Acknowledgements

We thank Timo Glatter, Jörg Kahnt, and Nataliya Teteneva for the support with the proteomics analysis, Gabriele Malengo for the support with microscopy and data analysis, and Sarah Hoch for the support with the immunoblot analysis. This work was supported by the Max Planck Society funding to VS, by the Hessian Ministry of Higher Education, Research, and the Arts (HMWK)–LOEWE research cluster "Diffusible Signals" subproject A1 grant to VS, by the National Research Foundation, Republic of Korea grant NRF-2019R1A6A3A03031885 to JWL, by the Swiss National Science Foundation grants 100.001.588 and NCCR Microbiomes 51NF40_180575 to WDH, by the Deutsche Forschungsgemeinschaft grant LA 4572/1-1 to LL, the National Institutes of Health grants NIH-NIAID- 5T32AI112541-08 to SLC and 1R21AI175788-01 and R01 AI168468-01 to MH.

## Author contributions

**Jae-Woo Lee**: Conceptualization; Data curation; Formal analysis; Funding acquisition; Investigation; Visualization; Methodology; Writing—original draft; Writing—review and editing. **Liyun Wang**: Conceptualization; Resources; Formal analysis; Investigation; Visualization; Methodology; Writing—review and editing. **Sarah L Comer**: Investigation; Methodology; Writing—review and editing. **Remy Colin**: Resources; Software; Formal analysis; Investigation; Methodology; Writing—review and editing. **Mollie M Gidney**: Investigation; Writing—review and editing. **Leanid Laganenka**: Formal analysis; Funding acquisition; Investigation; Visualization; Methodology; Writing—original draft; Writing—review and editing. **Wolf-Dietrich Hardt**: Supervision; Funding acquisition; Project administration; Writing—review and editing. **Maria Hadjifrangiskou**: Conceptualization; Supervision; Funding acquisition; Writing—original draft; Project administration; Writing—review and editing. **Victor Sourjik**: Conceptualization; Supervision; Funding acquisition; Writing—original draft; Project administration; Writing—review and editing.

Source data underlying figure panels in this paper may have individual authorship assigned. Where available, figure panel/source data authorship is listed in the following database record: biostudies:S-SCDT-10_1038-S44318-025-00595-x.

## Funding

## Disclosure and competing interests statement

The authors declare no competing interests.

# Expanded View Figures

**Figure EV1.   Effect of Tls on flagellar gene expression and motility in *E. coli* MG1655.**

(A) Fluorescence levels of P*fliC-gfp* normalized by $OD_{600}$ values in MG1655 carrying either empty vector pTrc99A or pTrc99A-Tls induced with indicated concentrations of IPTG. Measurements were performed in the log phase of growth in a plate reader. Promoter region of *gadX*, RpoS-dependent gene unrelated to motility, was used as a control. Values represent the means and standard deviations of a minimum of three biological replicates, in each case normalized to the reporter activity in the reference strain (MG1655/pTrc99A). Statistical significance was determined using unpaired two-tailed Student's *t* test. The *P* values are denoted as ns ($P > 0.05$), * ($P < 0.05$), ** ($P < 0.005$), *** ($P < 0.001$). (*P* values from left to right: ***$P = 8.88E-06$, ***$P = 3.35E-09$, ***$P = 1.59E-06$, ***$P = 3.60E-09$, ***$P = 2.52E-11$, ***$P = 4.71E-12$, ns $= 0.0721$). (B–E) Fraction of swimmers (B), swimming speed (C), chemotactic velocity (D), and chemotactic bias (E) for MG1655 carrying either empty vector pTrc99A or pTrc99A-Tls induced by indicated concentrations (B, C) or 50 µM (D, E) of IPTG. See Methods for details. Values represent the means and standard deviations of three biological replicates. Statistical significance was determined using unpaired two-tailed Student's *t* test (*P* values from left to right: ns $= 0.36$, *$P = 0.0067$, ***$P = 0.0003$ (B); ns $= 0.1903$, **$P = 0.0029$, ***$P = 0.0002$ (C); ns $= 0.12$ (D); ns $= 0.74$ (E). Source data are available online for this figure.

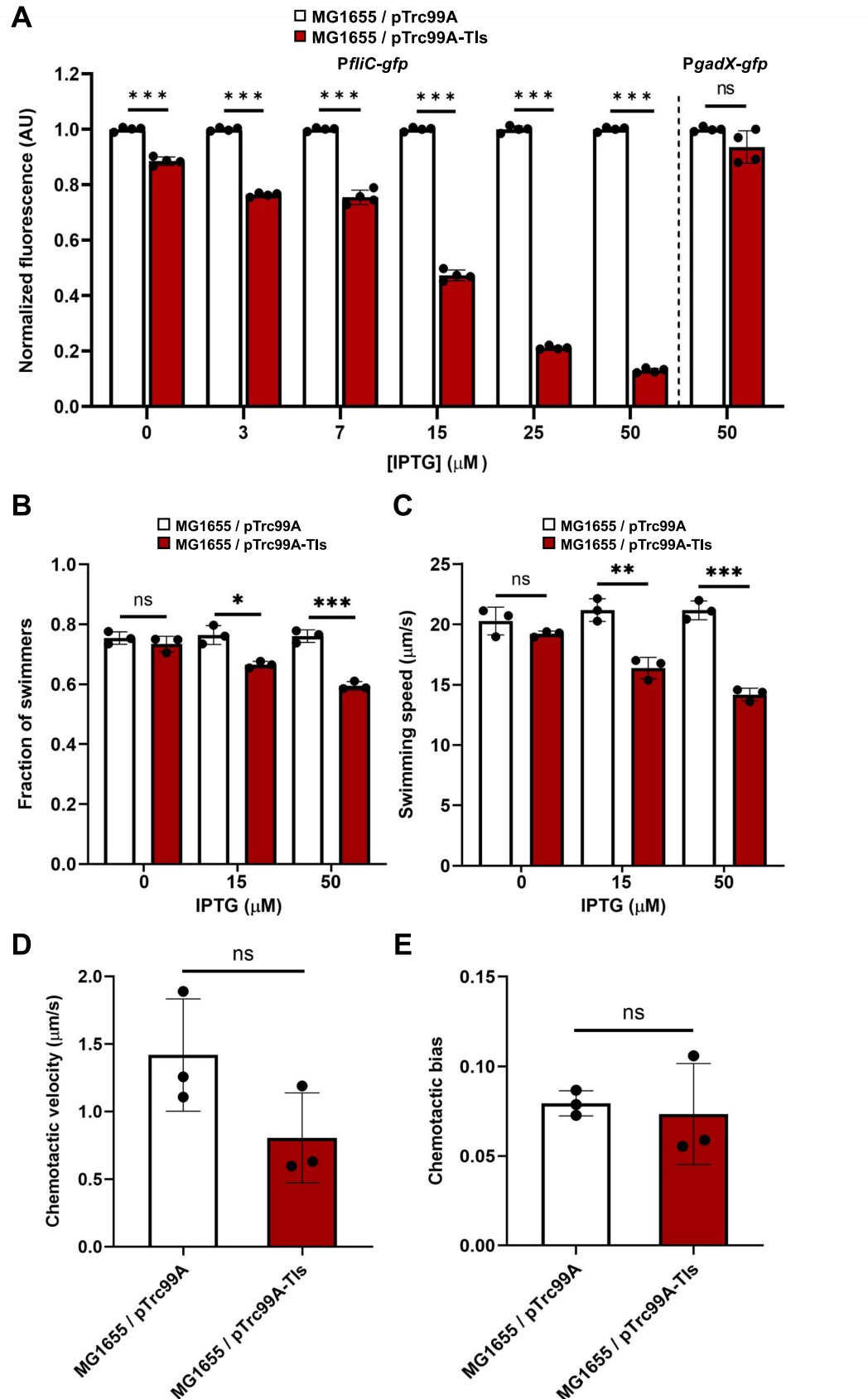

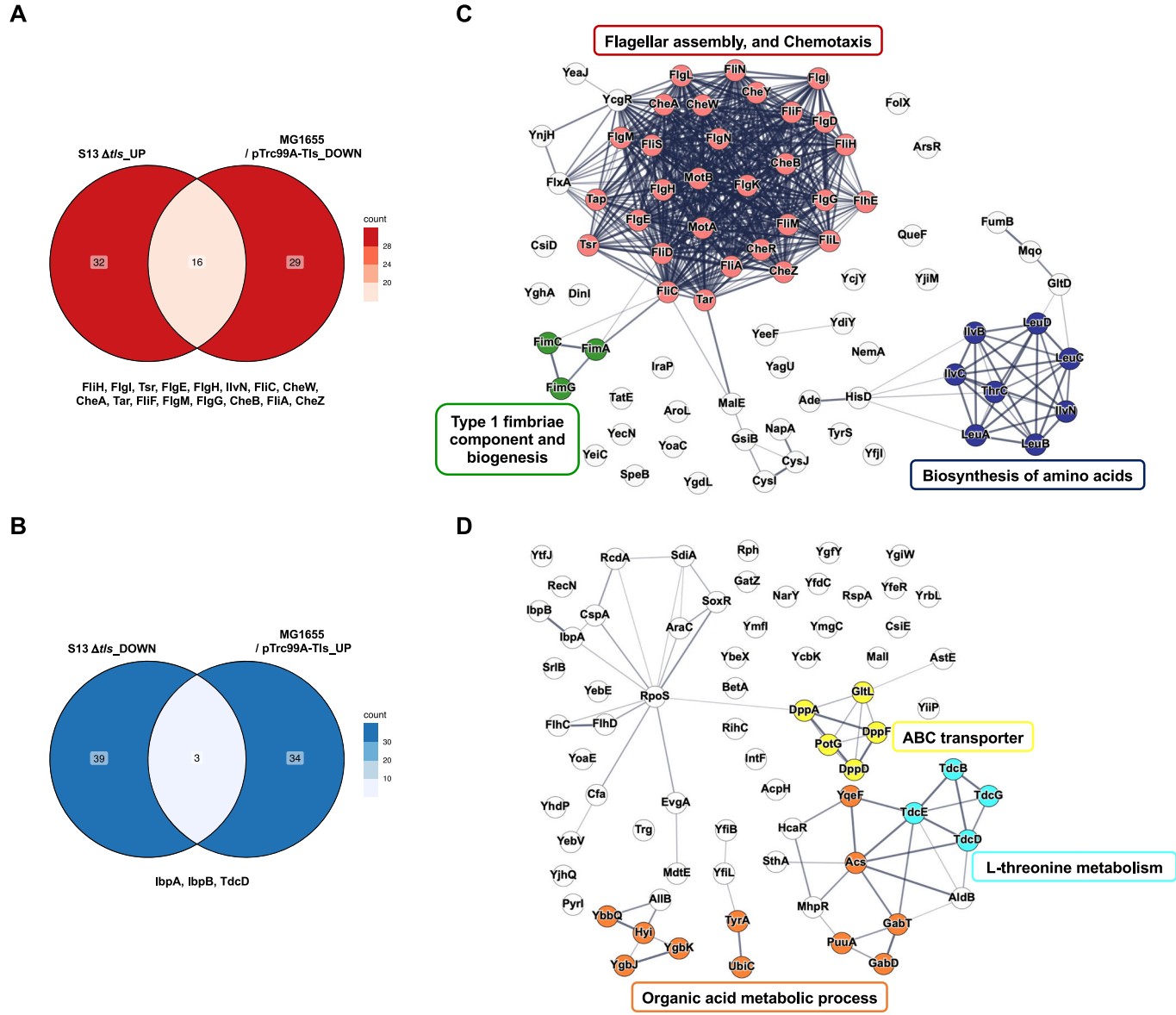

**Figure EV2.** Downregulation of class 2 and class 3 flagellar proteins by Tls.

(A) Venn diagram of proteins upregulated in *E. coli* S13 upon deletion of *tls* and downregulated in *E. coli* MG1655 upon expression of Tls from pTrc99A-Tls at 15 µM IPTG induction. (B) Venn diagram of proteins downregulated in *E. coli* S13 upon deletion of *tls* and upregulated in *E. coli* MG1655 upon expression of Tls from pTrc99A-Tls at 15 µM IPTG induction. The protein groups shared by both strains are shown below of each Venn diagrams, and all proteins identified in both strains are listed in Appendix Tables S2–S5. (C, D) The STRING diagram of the clustering of proteins downregulated (C) or upregulated (D) by Tls in one or both strains. For the downregulated proteins, the clusters of flagellar assembly and chemotaxis, biosynthesis of amino acids, and type 1 fimbriae component and biogenesis are highlighted in red, blue, and green, respectively. For the upregulated proteins, the clusters of ABC transporter, L-threonine metabolism, and organic acid metabolic process are highlighted in yellow, cyan, and orange, respectively. The thickness of the lines indicates the strength of data support in STRING.

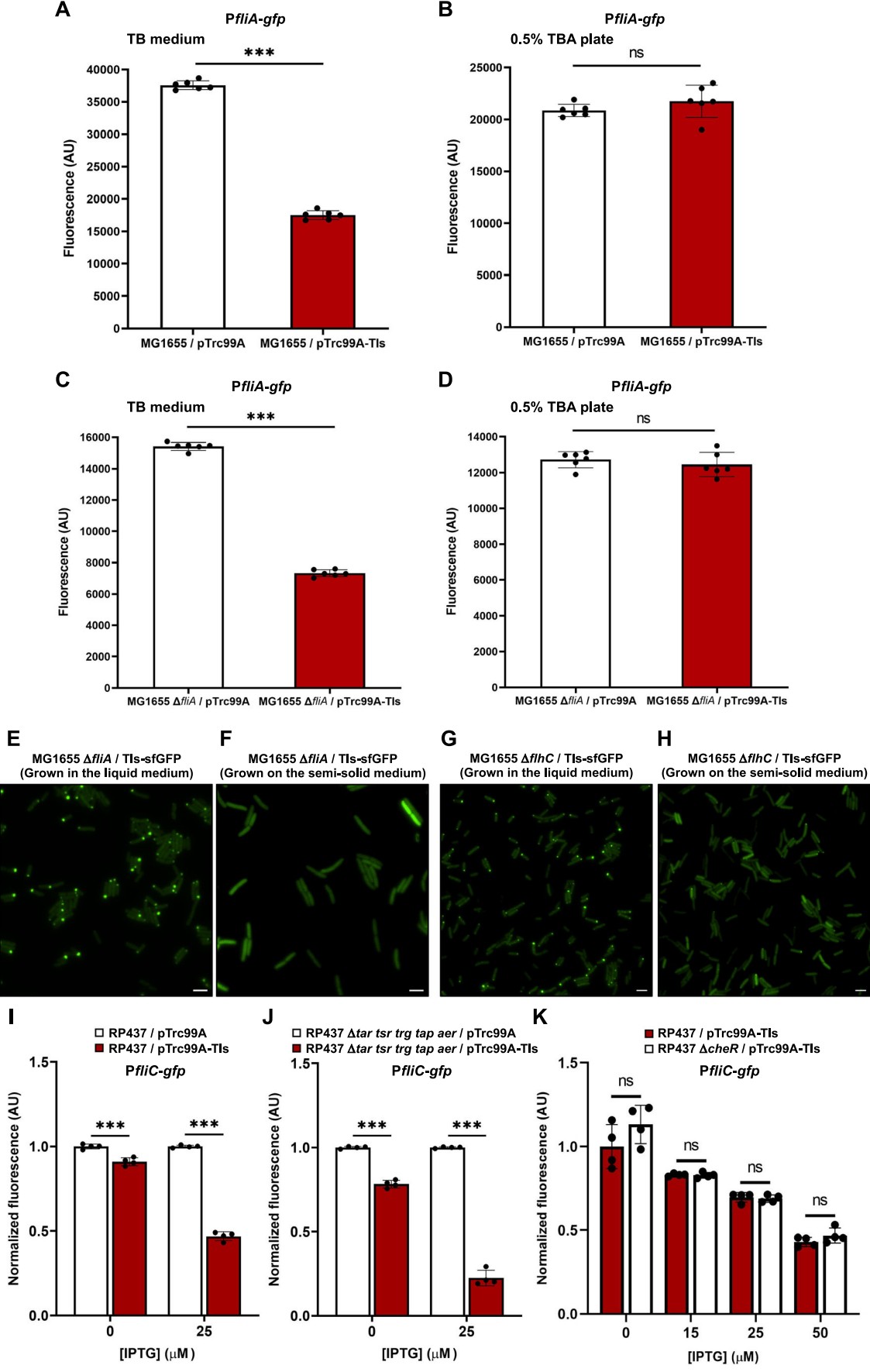

◀

**Figure EV3. Surface-sensitive regulation of flagellar gene expression by Tls in chemotaxis components-independent manner.**

(A–D) Fluorescence levels of P*fliA-gfp* at log phase normalized by $OD_{600}$ in MG1655 (**A, B**) and MG1655 Δ*fliA* (**C, D**) cells carrying either empty pTrc99A vector or pTrc99A-Tls expression plasmid induced with 15 μM IPTG, grown in liquid TB medium (**A, C**) or on the surface of 0.5% TB agar (TBA) plate (**B, D**). Values represent the means and standard deviations of a minimum of three biological replicates. Statistical significance was determined using unpaired two-tailed Student's *t* test. The *P* values are denoted as ns ($P > 0.05$), * ($P < 0.05$), ** ($P < 0.005$), *** ($P < 0.001$). (***$P = 2.00E\text{-}13$ (**A**); ns $= 0.2208$ (**B**); ***$P = 4.90E\text{-}14$ (**C**); ns $= 0.43$ (**D**)). (**E-H**) Cellular localization of Tls-sfGFP in MG1655 Δ*fliA* (**E, F**) or MG1655 Δ*flhC* (**G** and **H**) cells grown either in the liquid TB medium (**E, G**) or on 0.5% TBA plate (**F, H**). Representative images are shown from three biological replicates. Scale bars, 3 μm. (**I–K**) Fluorescence levels of P*fliC-gfp* normalized by $OD_{600}$ values in RP437 (**I**) and its receptor-less (**J**) or *cheR* (**K**) mutant carrying either empty vector pTrc99A or pTrc99A-Tls induced with indicated concentrations of IPTG. Measurements were performed in the log phase of growth in a plate reader. Values represent the means and standard deviations of a minimum of three biological replicates, in each case normalized to the reporter activity in the reference strain (RP437/pTrc99A, RP437 Δ*tar tsr trg tap aer*/pTrc99A or RP437/pTrc99A-Tls). Statistical significance was determined using unpaired two-tailed Student's *t* test (*P* values from left to right: ***$P = 0.0005$, ***$P = 1.90E\text{-}08$ (**I**); ***$P = 1.50E\text{-}06$, ***$P = 4.60E\text{-}08$ (**J**); ns $= 0.18$, ns $= 0.78$, ns $= 0.67$, ns $= 0.20$ (**K**)). Source data are available online for this figure.

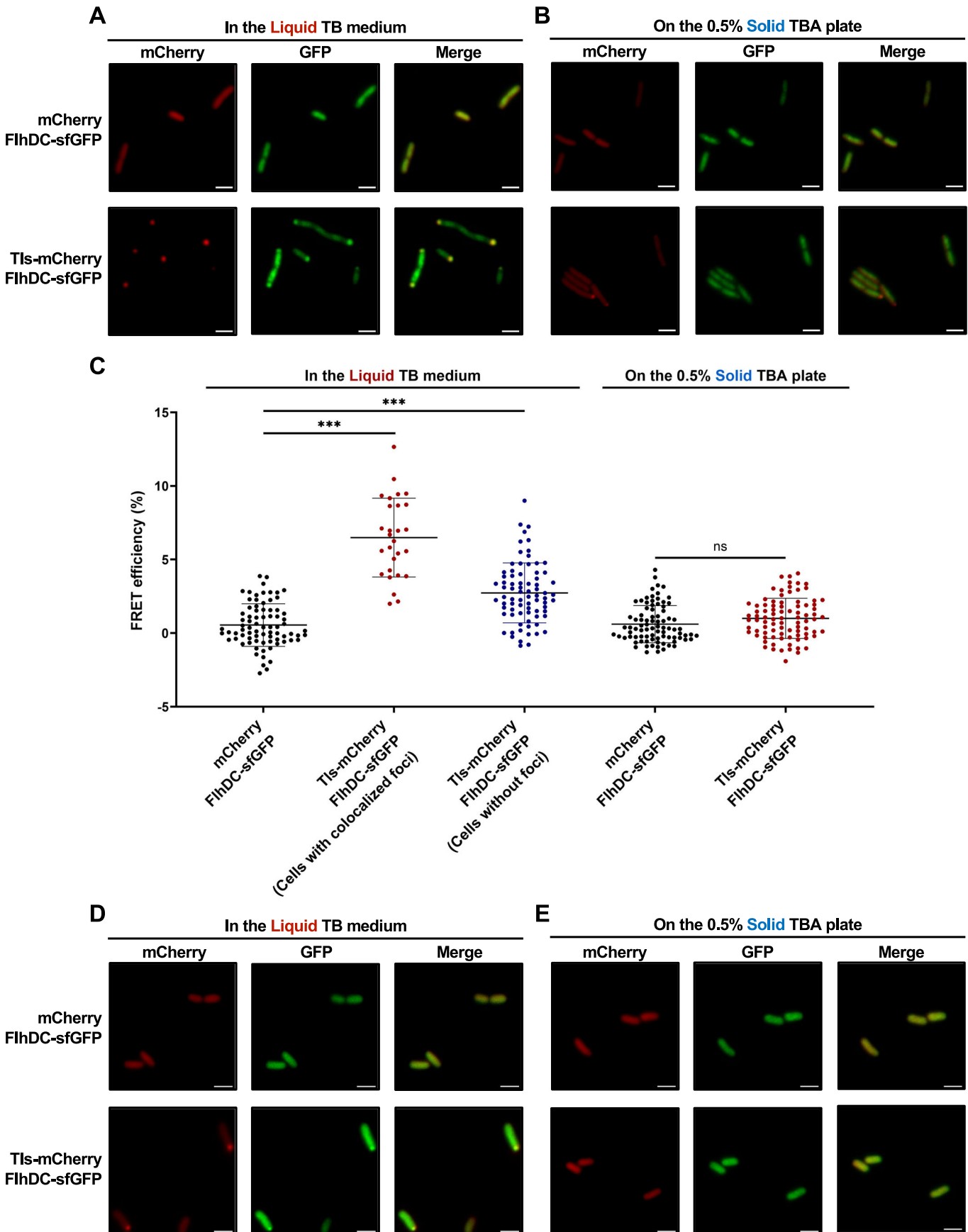

◀ **Figure EV4. Interaction of Tls with FlhDC.**

(A, B) Localization of Tls-mCherry or mCherry alone and FlhDC-sfGFP in MG1655 cells grown either in liquid TB medium (A) or on 0.5% TBA plate (B), as described in Fig. 6F. The mCherry channel, GFP channel, and merged images are shown. Representative images are shown from three biological replicates. Scale bars, 2 μm. (C) FRET measurements of complex formation between FlhDC-sfGFP and Tls-mCherry, co-expressed in S13 cells grown either in liquid TB medium or on 0.5% TBA plate, as indicated. Co-expressed mCherry was used as a negative control. FlhDC-sfGFP expression was induced with 0.002% L-arabinose in liquid TB and with 0.004% on 0.5% TBA plate. Tls-mCherry expression was respectively induced with 15 μM or 30 μM IPTG and mCherry expression was respectively induced with 40 μM or 200 μM IPTG. FRET efficiency was determined by acceptor photobleaching in individual cells as described in the Methods. Symbols represent the FRET values in individual cells, measured in three biological replicates (7–33 cells per replicate), with the means and standard deviations being indicated. For liquid-grown cells, the distinction was made between cells that exhibit colocalized foci of Tls-mCherry and FlhDC-sfGFP and those that do not. No colocalized foci were observed in surface-grown cells. Statistical significance was determined using unpaired two-tailed Student's $t$ test. The $P$ values are denoted as ns ($P > 0.05$), * ($P < 0.05$), ** ($P < 0.005$), *** ($P < 0.001$). ($P$ values from left to right: ***$P = 6.76E\text{-}27$, ***$P = 1.41E\text{-}12$, ns $= 0.05$). (D, E) Localization of mCherry only, or of Tls-mCherry and FlhDC-sfGFP in S13 cells grown either in liquid TB medium (D) or on 0.5% TBA plate (E), as described in (C). The mCherry channel, GFP channel, and merged images are shown. Representative images are shown from two biological replicates. Scale bars, 2 μm. Source data are available online for this figure.

**A**

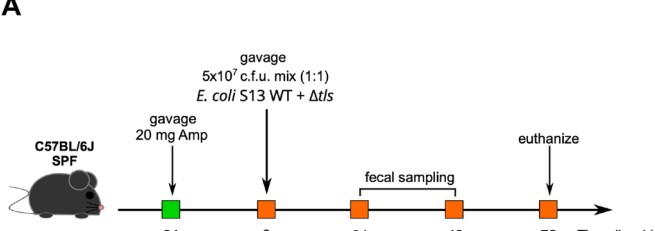

**B**

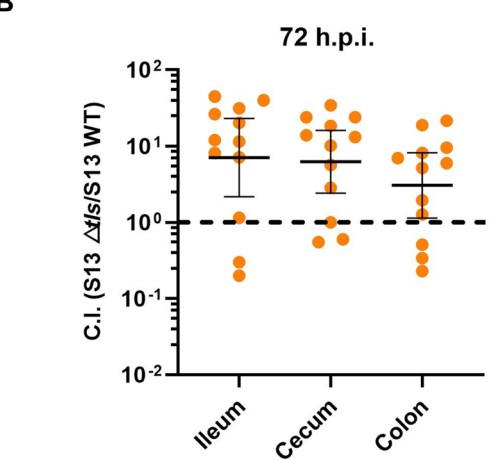

**Figure EV5. Testing role of Tls in gut proliferation.**

(A) Experimental scheme of competitive oral inoculation. C57BL/6J SPF mice were pre-treated with 20 mg of ampicillin by oral gavage 24 h before infection with *E. coli* (1:1 mix of S13 WT and Δ*tls* strains). Feces were collected at 24 and 48 h.p.i., and mice were euthanized at 72 h.p.i. (B) The CI values of Δ*tls* in Δ*tls*/WT competitive infection along the gut at 72 h.p.i. Analyses were performed using 12 mice (individual dots). The line depicts the geometric mean +/− 95% CI. The dashed line indicates the CI value of 1. Source data are available online for this figure.

