## [Peer Review File · The EMBO Journal]

Control of flagellar gene expression by a chemotaxis receptor-like regulator in pathogenic *Escherichia coli*

Jae-Woo Lee, Liyun Wang, Sarah Comer, Remy Colin, Mollie Gidney, Leanid Laganenka, Wolf-Dietrich Hardt, Maria Hadjifrangiskou, and Victor Sourjik

Corresponding author: Victor Sourjik (victor.sourjik@mpi-marburg.mpg.de)

Review Timeline:

Submission Date:	12th Nov 24
Editorial Decision:	13th Jan 25
Revision Received:	7th Jun 25
Editorial Decision:	11th Jul 25
Revision Received:	12th Sep 25
Accepted:	29th Sep 25

Editor: Ieva Gailite

Transaction Report:

Dear Victor,

Thank you for submitting your manuscript for consideration by the EMBO Journal. We have now received comments from three reviewers, which are included below for your information.

As you can see, all reviewers are generally positive in their assessment and find the proposed role of TIs in FlhDC regulation per se of interest. However, they also raise a range of concerns that would need to be addressed before they can support publication. In particular, they request further analysis of TIs interaction with FlhDC and strengthening of the evidence for its role in chemotaxis-independent, surface contact-dependent motility regulation. From my side, I find the raised points generally reasonable. I therefore invite you to address these remaining comments in a revised manuscript. I think that it would be useful to discuss the revision in more detail via email or phone/videoconferencing - please let me know which option you prefer.

We generally allow three months as standard revision time, which can be extended to six months in the case of major revisions. Should you foresee a problem in meeting this deadline, please let us know in advance to discuss an extension. As a matter of policy, competing manuscripts published during this period will not negatively impact on our assessment of the conceptual advance presented by your study. However, please contact me as soon as possible upon publication of any related work to discuss the appropriate course of action.

When preparing your letter of response to the referees' comments, please bear in mind that this will form part of the Review Process File and will therefore be available online to the community. For more details on our Transparent Editorial Process, please visit our website: <https://www.embopress.org/page/journal/14602075/authorguide#transparentprocess>. Please also see the attached instructions for further guidelines on preparation of the revised manuscript.

Please feel free to contact me if have any further questions regarding the revision. Thank you for the opportunity to consider your work for publication, and I look forward to discussing your revision with you.

With best regards,

leva

leva Gailite, PhD
Senior Scientific Editor
The EMBO Journal
Meyerhofstrasse 1
D-69117 Heidelberg
Tel: +4962218891309
i.gailite@embojournal.org

- a point-by-point response to the referees' comments, with a detailed description of the changes made (as a word file).
- a word file of the manuscript text.
- individual production quality figure files (one file per figure)
- a complete author checklist, which you can download from our author guidelines

(<https://www.embopress.org/page/journal/14602075/authorguide>).

- Expanded View files (replacing Supplementary Information)

- a Reagents and Tools Table as part of the Methods section, which can be downloaded from our author guidelines

(<https://www.embopress.org/page/journal/14602075/authorguide#structuredmethods>)

We realize that it is difficult to revise to a specific deadline. In the interest of protecting the conceptual advance provided by the work, we recommend a revision within 3 months (13th Apr 2025). Please discuss the revision progress ahead of this time with the editor if you require more time to complete the revisions.

Referee #1:

This paper by Lee and colleagues describes the regulatory role of TIs in the expression of flagellar and chemotaxis genes. TIs is a protein derived from a truncation of the Trg chemoreceptor. While Trg is widely found in enteric bacteria, TIs is present only in a restricted group of extraintestinal pathogenic enterobacteria. The authors show that, in response to a mechanical cue such as a transition from liquid to solid media, TIs relocates from the cytoplasm to the cell poles. At this location, TIs sequesters the hetero-oligomeric transcriptional regulator FlhDC, thereby inhibiting the transcription of flagellar and chemotaxis genes. The authors also provide evidence that the cellular localization of TIs does not depend on other chemoreceptors (the determinants of its localization remain elusive) and that the TIs conserved N-terminal tail is essential for TIs-mediated regulation. Finally, they show that TIs enhances attachment to urinary tract cells in uropathogenic *E. coli* while inhibiting migration and proliferation in the murine gut, providing evidence of the importance of TIs during infection. The manuscript is well-written, the results are original, and the experiments have been performed rigorously. The findings are also of significant impact in the fields of chemotaxis, bacterial motility, and pathogenic enterobacterial infections. My only concern lies in the absence of few experiments that could further support the molecular mechanisms underlying TIs regulation.

Major Comments

While the cell biology data suggests that TIs regulates the transcription of flagellar and chemotaxis genes by sequestering FlhDC from the nucleoid, experimental evidences are missing:

- Does TIs directly interact with FlhDC?
- Are the TIs-mCherry and FlhDC-GFP fusions functional?
- Figure 7A: The amount of FlhDC sequestered at the pole appears small compared to the strong effect on transcription. How would the authors explain this discrepancy?
- How is the transcription of PflhC-GFP impacted by the K42E mutation in TIs?
- Why does TIs not interact with FlhDC when diffused in the cytoplasm on solid media? Could instead a TIs-FlhDC complex form that regulates transcription?

Figures 3 and 5: Why are the effects of TIs deletion and overexpression compared in different backgrounds? If S13/pTrc1655-TIs is available, couldn't the results be merged into the same plots?

Page 5 and Figure 2A: While the growth defect of Δ tIs compared to the wild type appears significant, it is quite mild and might not fully explain the motility defect on soft agar. Could there instead be a post-transcriptional inhibition of motility on agar?

Does the overexpression of TIs lead to growth defects?

How does TIs-GFP localize when cells are grown on the 0.27% agar used for Figure 2A?

Minor Comments

Does TIs contain methylation sites?

Figure 2B: Specify what the black and white curves represent.

Page 9: It would be helpful to test the localization of TIs in Δ mcp mutants to establish a direct correlation.

Referee #2:

In this manuscript, the authors investigate a truncated version of a chemoreceptor called Tls, which they find broadly conserved in a specific *E. coli* phylogroup. They demonstrate that a mutant lacking Tls displays an increased fraction of swimming cells in liquid media and increased expression of flagellar genes. Intriguingly, expression of *tsl* in an *E. coli* K12 lab strain also reduced flagellar gene expression, indicating that the underlying regulatory mechanism is conserved. The authors show data to suggest that Tsl-mediated repression of motility is specific for planktonic cells but that this response is quenched in surface exposed *E. coli* cells. This, and the observation that Tsl localizes to the poles of cells grown in liquid media, but is delocalized in cells grown on semi-solid agar, led them to suggest that subcellular positioning of Tsl is linked to flagellar gene expression control. A possible mechanism for how Tsl regulates flagellar gene expression is offered by the observation that in the presence of Tsl, the master flagellar regulator FlhDC localizes to the cell poles.

The authors then identify a conserved region of 9 amino acids at the N-terminus of Tsl, which is required and sufficient for Tsl-mediated motility control. In particular, grafting this peptide onto the fluorescent reporter protein mCherry resulted in its polar localization and also partially restored flagellar repression of a *tsl* mutant. Polar localization of Tsl was not dependent on other chemoreceptors, excluding receptor clustering as a mechanism of control. Finally, animal studies revealed no evidence for a role of Tsl in the infection of the urinary tract, but exposed a competitive advantage during intestinal colonization of a gavage mouse model.

The authors present highly interesting and novel information on a variant chemoreceptor and its potential adaptation to surface sensing and motility control in *E. coli*. Based on this, they present an attractive model in which Tsl changes its polar localization depending on surface recognition and by sequestering FlhDC to polar sites, specifically alters motility and *fla/che* gene expression. However, several aspects of the model remain speculative and require additional experimental support to make the case stronger. Also, several points need to be clarified to make this work more comprehensible for the reader. Below is list of points that require attention.

Major points:

- The authors propose an interesting model suggesting that Tsl-mediated polar sequestration of FlhDC is responsible for flagellar gene expression and motility control. The idea is that the available cytoplasmic pool of FlhDC becomes limiting under conditions where polar Tsl sequesters a fraction of FlhDC away from the cytoplasm. At this point the model is largely based on indirect data (protein localization; gene expression). Additional experiments are needed in support of this idea, e.g., demonstrating that Tsl directly interacts with FlhDC (via its N-terminal sequence) could add substantial weight to the model.
- It is difficult to reconcile how a 9-aa peptide can be responsible for Tsl tagging to the cell pole and, at the same time, recruit FlhDC to the same compartment. This is largely based on the surprising observation that grafting this short sequence onto a fluorophore partially restored flagellar repression in a *tsl* mutant background. Does peptide grafting require the acceptor protein to be a dimer? It was recently shown that mCherry is dimeric in *E. coli* (doi: <https://doi.org/10.1101/2024.03.28.587235>). Does the grafting experiment also work with a truly monomeric fluorophore or a monomeric alternative acceptor protein?
- on p.5 the authors claim that the increased fraction of motile cells in a *tsl* mutant are '...due to its negative effect on flagellar gene expression in liquid'. While there is clearly a correlation, these data do not show a causal link between *fla* gene expression and motility. Please phrase more carefully or add additional data to support this point.
- Along the same lines, does the more limited increase in *fla* gene expression in UT189 (as compared to S13) scale with a more limited increase of the number of motile cells? Similarly, does the gradual reduction of *fla* gene expression with increasing *tsl* expression levels (Fig. 2E) scale with the fraction of motile S13 bacteria? And finally, does Tsl-mediated reduction of *fla* gene expression (Fig. EV3D) scale with numbers of motile cells in MG1655?
- The authors show that expression of a TlsN-mCherry fusion protein reduces *fla* gene expression. Does it also influence motility, i.e., numbers of motile cells?
- In Figs. 3C,D and EV4 the authors show protein expression upon deletion and overexpression of *tsl*, respectively. These experiments were carried out in 2 different strains of *E. coli*. Does overexpression of *tsl* in S13 and MG1655 produce similar (*fla/che*-specific) results?
- Capping the conserved N-terminal peptide renders Tsl unable to repress flagellar gene expression (Fig. 4). How stable is this mutant variant?
- Tsl localization studies in liquid and on surfaces were carried out with MG1655. Does Tsl also localize in S13 and is polar localization of Tsl dependent on its expression from a plasmid?
- The authors show that Tls switches its localization upon transition from liquid to solid substrate or vice versa within 30-60 hours. It would be interesting to address if gene expression and motility follow a similar timing. Such experiments would help strengthening the connection between Tsl localization, gene expression and cellular behavior.
- The authors original observations are that a *tsl* mutant shows a higher fraction of motile cells and that Tsl localization is influenced by the surface/liquid environment. While these are single cell observations, the authors do not follow up the link to gene expression at the single cell level. Scoring Tsl (or TlsN-mCherry) localization and gene expression in the same cells under different conditions would provide a powerful experimental link between individual cell behavior and FlhDC-mediated

transcriptional control.

- For instance, the idea that *E. coli* induces its *fla* and *che* genes specifically on surfaces is counterintuitive as one would rather expect downregulation of motility genes during surface adaptation. Can the authors provide some model or rationale for the reader to be able to grasp the physiological context or the general idea behind their observations? In general, I find the discussion rather uninspiring and superficial. E.g., it would be highly interesting for the reader if the authors would be discussing the FlhDC sequestration model in more detail and deliberate on how it could be further scrutinized.
- The authors show that FlhDC localizes to the cell poles in a Tsl-dependent manner (Fig. 7). However, polar foci are very weak, FlhDC levels seem to fluctuate quite a bit from cell to cell and co-localization experiments that could help clarify what is going on and strengthen a direct role of Tsl in FlhDC recruitment are missing. Furthermore, these experiments were carried out in the *E. coli* lab strain and should ideally be repeated in *E. coli* S13.

Minor points:

- To better illustrate the truncation of Tsl it would be instructive to show the full-length structure of Trg in Fig. 1B with its periplasmic, membrane-spanning and cytoplasmic parts clearly marked.
- Please add a simple control experiment excluding that plasmid-based expression of Tsl (Fig. 2E) affects *E. coli* growth.
- p.6: 'Interestingly, the abundance of FlhD and FlhC even increased in this latter case (Fig. 3D).' Was this unexpected? Please comment.
- Can the authors comment on the non-flagellar or chemotaxis proteins that are up- or down-regulated in response to Tsl availability (metabolism, Type 1 pili...)? What could be a possible link to FlhDC?
- The first two paragraphs of the discussion are more like an introduction and could be moved to the intro section.

Referee #3:

This work examines a unique chemoreceptor-like protein, Tls, which is widely present in the B2 phylogroup of *Escherichia coli*. The authors found that this protein controls bacterial motility by repressing flagellar gene expression through recruitment of FlhDC to polar foci, sequestering it away from DNA. Notably, the polar localization of Tls and its mediated inhibition of flagellar gene expression only occur during growth in liquid medium, not in porous medium or on surfaces. The N-terminal fragment of Tls retains similar functionality as the intact protein when fused to mCherry. The authors demonstrated the effect of Tls on motility regulation in both bladder and gut during infection.

Overall, Lee et al. showed that the Tls protein represses flagellar gene expression, with this repression closely linked to both Tls polar foci formation and the cellular growth environment. They also demonstrated that the TlsN-mCherry fusion functions similarly to intact Tls, showing both gene repression activity and polar localization. While most results are convincing, the mechanism and physiological significance require further explanation. Here are specific points:

1. The authors' conclusion about Tls not affecting chemotaxis, based on comparing S13 and S13 Δ tls motility on TBSA, needs additional validation. Multiple factors influence TBSA assays, including growth, motility, and chemotaxis. Additional controls are needed, such as chemotaxis-defective strains, or direct chemotaxis assays using ibidi channels.
2. The authors report improved motility of S13 Δ tls cells in liquid TB medium, with more swimmers but unchanged swimming speed and reorientation frequency. Two issues arise: First, the trajectories in Figure 2B show typical surface-associated circular swimming, making it unclear how reorientation frequency was determined. Second, for surface-located trajectories, it's important to distinguish whether non-motile bacteria are surface-attached or undergoing Brownian motion. Surface attachment would suggest Tls affects adhesion protein secretion, while Brownian motion would indicate effects on motor or flagellar filament synthesis.
3. The study should include motility measurements in tethering buffer, which is standard for motility and chemotaxis research.
4. The reported Tls-mediated downregulation of flagellar and chemotaxis proteins in S13 and MG1655 seems inconsistent with minimal effects on swimming speeds and frequency of cell reorientations in liquid. This discrepancy requires explanation.
5. The fourth and sixth Results sections both address N-terminal region function and probably should be combined. While TlsN-mCherry functions similarly to Tls in both S13 and MG1655 strains by repressing the flagellar genes, using S13 for localization studies would be more appropriate given its natural expression of Tls. Additional questions include whether TlsN functions without fusion partners and whether the E2348/69 Tls variant shows similar activity.
6. While the authors show that surface-dependent localization and flagellar gene repression do not require the chemotaxis signaling system, more discussion is needed on how Tls or TlsN aggregate at cell poles during liquid growth. The authors mention biomolecular condensates. Even if that was the case, why they appear at cell poles?

7. The FlhDC recruitment analysis in MG1655 reveals weak polar aggregation of non-mutated FlhDC, with enhanced aggregation after mutation. This may reflect insufficient recruitment capacity of non-native TIs to overcome DNA binding. Using the S13 strain would provide more convincing results. Additionally, colocalization studies of TIs and FlhDC could clarify their interaction at cell poles.

Referee #1:

This paper by Lee and colleagues describes the regulatory role of TIs in the expression of flagellar and chemotaxis genes. TIs is a protein derived from a truncation of the Trg chemoreceptor. While Trg is widely found in enteric bacteria, TIs is present only in a restricted group of extraintestinal pathogenic enterobacteria. The authors show that, in response to a mechanical cue such as a transition from liquid to solid media, TIs relocalizes from the cytoplasm to the cell poles. At this location, TIs sequesters the hetero-oligomeric transcriptional regulator FlhDC, thereby inhibiting the transcription of flagellar and chemotaxis genes.

The authors also provide evidence that the cellular localization of TIs does not depend on other chemoreceptors (the determinants of its localization remain elusive) and that the TIs conserved N-terminal tail is essential for TIs-mediated regulation. Finally, they show that TIs enhances attachment to urinary tract cells in uropathogenic *E. coli* while inhibiting migration and proliferation in the murine gut, providing evidence of the importance of TIs during infection.

The manuscript is well-written, the results are original, and the experiments have been performed rigorously. The findings are also of significant impact in the fields of chemotaxis, bacterial motility, and pathogenic enterobacterial infections. My only concern lies in the absence of few experiments that could further support the molecular mechanisms underlying TIs regulation.

We thank the Reviewer for acknowledging the originality and quality of our manuscript and its potential impact in the field and for the suggestions on manuscript improvement. We have now addressed all the points raised by the Reviewer.

Major Comments

While the cell biology data suggests that TIs regulates the transcription of flagellar and chemotaxis genes by sequestering FlhDC from the nucleoid, experimental evidences are missing:

- Does TIs directly interact with FlhDC?

We have now performed measurements to test the interaction between TIs-mCherry and FlhDC-GFP in the cell using FRET. Consistent with our model, these data confirms that TIs and FlhDC are part of the same complex when bacteria are grown in liquid but not when the grown is on surface. The results are now shown in Figure 6F,G and in Expanded View Figure 4 and discussed in the text (lines 305-319). The biochemical characterization of this complex formation and of its regulation will be the subject of our future work, but its dependence on growth conditions further suggests that it is either modulated or mediated by other cellular components. It was mentioned already briefly in the previous version of our manuscript, and we now discuss it in greater detail in the Discussion section (lines 371-379).

- Are the TIs-mCherry and FlhDC-GFP fusions functional?

We now show the data confirming that TIs-mCherry can repress flagellar gene expression in Figure 5B. FlhDC-GFP fusion cannot activate expression of flagellar genes, which is not uncommon for fluorescent protein fusions to transcription factors. However, it shows localization to the nucleoid (Figure 6A), suggesting that its DNA-binding function is preserved. We now mention it in the Results when introducing this fusion (lines 284-287).

- Figure 7A: The amount of FlhDC sequestered at the pole appears small compared to the strong effect on transcription. How would the authors explain this discrepancy?

We thank the Reviewer for raising this point. The aim of this experiment was to qualitatively show that TIs can sequester FlhDC to the complex it forms, which is now further supported by FRET data and the corresponding co-localization results, shown in Figure 6F,G and in Expanded View Figure 4 (see above). The quantitative discrepancy mentioned by the Reviewer might be due to different reasons, such as the overproduction of FlhDC-GFP above the native levels of FlhDC in our microscopy experiments, but possibly also due to the impact of GFP fusion on the ability of FlhDC to be recruited. Even more likely is that not all of the sequestered inactive FlhDC must localize to the pole but a substantial fraction of this complex might be present as smaller cytoplasmic complexes. This is now mentioned in the text (lines 288-295 in the Results and lines 371-379 in the Discussion).

- How is the transcription of PflhC-GFP impacted by the K42E mutation in TIs?

We assume that the Reviewer refers here to the K42E K45E mutant of FlhC. It was mentioned already in the Results section when introducing this fusion that these mutations are known to disrupt DNA binding (and thus function of FlhDC) and the corresponding reference was cited, but we now clarify it in greater detail (lines 297-302).

- Why does TIs not interact with FlhDC when diffused in the cytoplasm on solid media? Could instead a TIs-FlhDC complex form that regulates transcription?

We thank the Reviewer for raising this interesting point. We specifically tested it now using FRET experiments, which suggested that the complex between TIs-mCherry and FlhDC-GFP is indeed not formed when cells are grown on solid medium. This provides additional argument in support of the hypothesis that the interaction between TIs and FlhDC may be dependent on (or mediated by) other cellular factors. This data is now shown in Figure 6F and discussed in the text (lines 305-312).

Figures 3 and 5: Why are the effects of TIs deletion and overexpression compared in different backgrounds? If S13/pTrc1655-TIs is available, couldn't the results be merged into the same plots?

We did not include the effect of TIs overproduction in S13 background in the initial version of the manuscript because the basal expression level of flagellar genes in the wildtype S13 strain is already low. But we agree with the Reviewer that the data could be added for consistency and show them now in the modified Appendix Figure 3E and mention it in the text (lines 131-132).

Page 5 and Figure 2A: While the growth defect of Δ TIs compared to the wild type appears significant, it is quite mild and might not fully explain the motility defect on soft agar. Could there instead be a post-transcriptional inhibition of motility on agar?

We think that this explanation is plausible, since the difference in spreading is also rather minor, and the difference in growth is reproducible (we now show additional data in Appendix Figure 3G. Nevertheless, we now rephrase it more as a hypothesis now (lines 104-111). In any case, this minor effect is the opposite to the regulation observed in liquid-grown cells, and the main point of showing this data was to emphasize the difference between motility in agar and in liquid.

Does the overexpression of TIs lead to growth defects?

We did not observe any significant growth defects of TIs expression in either S13 or MG1655 background, even at high induction. We agree with the Reviewer that these data are informative and show them now in Appendix Figure 3F,G and mention in the text (lines 142-144).

How does TIs-GFP localize when cells are grown on the 0.27% agar used for Figure 2A?

We tested this and now show that TIs-GFP localization is much reduced in cells grown on TBSA plates, consistent with our explanation. We now show this data (Appendix Figure 7C) and discuss them (lines 209-210).

Minor Comments

Does TIs contain methylation sites?

Yes, TIs contains two putative methylation sites (glutamates). They are now marked in our structural model of TIs in Figure 1B. However, we assume that the regulation of its activity by methylation is not important because TIs is still functional in the absence of the entire chemotaxis pathway (Expanded View Figure 3). We have now performed additional experiments to show that the function of TIs is not affected by deletion of the methyltransferase CheR (Expanded View Figure 3K).

Figure 2B: Specify what the black and white curves represent.

We apologize for the unclarity. This is now mentioned in the figure legend (lines 992-994).

Page 9: It would be helpful to test the localization of TIs in Δmcp mutants to establish a direct correlation.

As mentioned above, we observed that TIs is functional and localizes in the absence of the entire chemotaxis pathway, including chemoreceptors (Expanded View Figure 3). We now show additional data confirming that TIs expression results in similar inhibition of *fliC* reporter in the wildtype and Δmcp strain (Expanded View Figure 3I,J), confirming that it functions independently of other receptors.

Referee #2:

In this manuscript, the authors investigate a truncated version of a chemoreceptor called TIs, which they find broadly conserved in a specific E. coli phylogroup. They demonstrate that a mutant lacking TIs displays an increased fraction of swimming cells in liquid media and increased expression of flagellar genes. Intriguingly, expression of *tsl* in an E. coli K12 lab strain also reduced flagellar gene expression, indicating that the underlying regulatory mechanism is conserved. The authors show data to suggest that Tsl-mediated repression of motility is specific for planktonic cells but that this response is quenched in surface exposed E. coli cells. This, and the observation that Tsl localizes to the poles of cells grown in liquid media, but is delocalized in cells grown on semi-solid agar, led them to suggest that subcellular

positioning of Tsl is linked to flagellar gene expression control. A possible mechanism for how Tsl regulates flagellar gene expression is offered by the observation that in the presence of Tsl, the master flagellar regulator FlhDC localizes to the cell poles.

The authors then identify a conserved region of 9 amino acids at the N-terminus of Tsl, which is required and sufficient for Tsl-mediated motility control. In particular, grafting this peptide onto the fluorescent reporter protein mCherry resulted in its polar localization and also partially restored flagellar repression of a *tsl* mutant. Polar localization of Tsl was not dependent on other chemoreceptors, excluding receptor clustering as a mechanism of control. Finally, animal studies revealed no evidence for a role of Tsl in the infection of the urinary tract, but exposed a competitive advantage during intestinal colonization of a gavage mouse model.

The authors present highly interesting and novel information on a variant chemoreceptor and its potential adaptation to surface sensing and motility control in *E. coli*. Based on this, they present an attractive model in which Tsl changes its polar localization depending on surface recognition and by sequestering FlhDC to polar sites, specifically alters motility and *fla/che* gene expression. However, several aspects of the model remain speculative and require additional experimental support to make the case stronger. Also, several points need to be clarified to make this work more comprehensible for the reader. Below is list of points that require attention.

We thank the Reviewer for finding our results highly interesting and for suggestions for additional experiments and improved discussion of our findings.

Major points:

- The authors propose an interesting model suggesting that Tsl-mediated polar sequestration of FlhDC is responsible for flagellar gene expression and motility control. The idea is that the available cytoplasmic pool of FlhDC becomes limiting under conditions where polar Tsl sequesters a fraction of FlhDC away from the cytoplasm. At this point the model is largely based on indirect data (protein localization; gene expression). Additional experiments are needed in support of this idea, e.g., demonstrating that Tsl directly interacts with FlhDC (via its N-terminal sequence) could add substantial weight to the model.

We have now performed FRET and co-localization experiments showing that Tsls-mCherry and FlhDC-GFP indeed interact (or are at least part of the same complex) in bacteria grown in liquid, but not on solid surface (Figure 6F,G and Expanded View Figure 4). This dependence indicates that the interaction between Tsls and FlhDC may be conditional and dependent on (or mediated by) other cellular factors. We now discuss these points in the revised manuscript (lines 305-319).

- It is difficult to reconcile how a 9-aa peptide can be responsible for Tsl tagging to the cell pole and, at the same time, recruit FlhDC to the same compartment. This is largely based on the surprising observation that grafting this short sequence onto a fluorophore partially restored flagellar repression in a *tsl* mutant background. Does peptide grafting require the acceptor protein to be a dimer? It was recently shown that mCherry is dimeric in *E. coli* (doi: <https://doi.org/10.1101/2024.03.28.587235>). Does the grafting experiment also work with a truly monomeric fluorophore or a monomeric alternative acceptor protein?

We thank the Reviewer for this insightful suggestion and for pointing us to this recent publication. Following the Reviewer's suggestion, we have now grafted TslN to either monomeric or dimeric sfGFP variants and see a clear difference in their localization. This

indicates that, as hypothesized by the Reviewer, the dimeric nature of the acceptor protein is important to mediate the polar localization of TlsN fusions. Since the full-length Tls is likely to be a dimer, as the canonical Trg receptor from which it evolved, this helps to explain the role of the remaining Tls fragment. We now show these data in Figure 5C and Appendix Figure 8G and discuss them in the text (lines 253-265).

- on p.5 the authors claim that the increased fraction of motile cells in a *tsl* mutant are '..due to its negative effect on flagellar gene expression in liquid'. While there is clearly a correlation, these data do not show a causal link between *fla* gene expression and motility. Please phrase more carefully or add additional data to support this point.

Along the same lines, does the more limited increase in *fla* gene expression in UT189 (as compared to S13) scale with a more limited increase of the number of motile cells? Similarly, does the gradual reduction of *fla* gene expression with increasing *tsl* expression levels (Fig. 2E) scale with the fraction of motile S13 bacteria? And finally, does Tsl-mediated reduction of *fla* gene expression (Fig. EV3D) scale with numbers of motile cells in MG1655?

As suggested by the Reviewer, we have now performed more detailed analysis of the effects of Tls deletion, as well as of its overexpression, on motility (modified Figure 2C-E and Appendix Figure 3B-D and Expanded View Figure 1B,C). In general, effects of Tls on motility are consistent with changes in gene expression, including dose-dependence of Tls-mediated repression (compare Expanded View Figure 1A with Expanded View Figure 1B,C) but are milder, particularly at higher gene expression where the dependence of motility on flagellar expression saturates (as was recently shown in Lisevich et al, 2025 that we now cite). As a consequence (and also due to relatively large culture-to-culture variability in average motility), it appears that significant effects of Tls on motility can only be observed for larger changes in expression. We discuss our data in its context (lines 139-142).

- The authors show that expression of a TlsN-mCherry fusion protein reduces *fla* gene expression. Does it also influence motility, i.e., numbers of motile cells?

We have tested this as well, and apparently the milder inhibition of flagellar gene expression by TlsN-mCherry is not sufficient to produce a significant inhibition of motility, for the reasons mentioned above. We now show and mention this result (Appendix Figure 8E,F). (lines 238-241).

- In Figs. 3C,D and EV4 the authors show protein expression upon deletion and overexpression of *tsl*, respectively. These experiments were carried out in 2 different strains of *E. coli*. Does overexpression of *tsl* in S13 and MG1655 produce similar (*fla*/*che*-specific) results?

*We have now added the results of Tls overexpression in S13 on flagellar gene expression in Appendix Figure 3E. This data confirms that Tls overexpression has similar effect in S13 background, as expected. The reason why we compare the effects of *tsl* deletion in S13 and Tls expression in MG1655 is because the latter does not encode Tls and the wildtype MG1655 is effectively equivalent to the *tsl* deletion strain. In contrast, S13 already natively expresses Tls, and we thus compare the wildtype S13 to *tsl* deletion strain. We thus believe that the comparisons between the effect of expression of Tls in MG1655 and its natural expression in S13 is justified. Since flagellar gene expression in S13 already largely repressed by the native Tls, its overexpression in the wildtype strain can only result in moderate additional reduction of expression. We now better explain this in the text (lines 131-138).*

- Capping the conserved N-terminal peptide renders Tsl unable to repress flagellar gene expression (Fig. 4). How stable is this mutant variant?

We thank the Reviewer for this question. We have now performed a Western blot analysis of protein levels for Tls Δ N-GFP fusion. We observed that, similar to the full-length fusion shown already in the previous version of the manuscript (Appendix Figure 7A), this fusion is stable. However, we discovered that its expression level is substantially lower at the same induction compared to the full-length fusion. Since such lower expression level might also be the case for the untagged Tls Δ N, we decided to rather remove the data for this truncated version of Tls from Figure 6 and Appendix Figure 8. They were not essential for drawing our conclusions, as we already show that the conserved N-terminal peptide is sufficient to mediate regulation (apparently when fused to a dimeric protein, see above), which is an even stronger statement.

- Tsl localization studies in liquid and on surfaces were carried out with MG1655. Does Tsl also localize in S13 and is polar localization of Tsl dependent on its expression from a plasmid?

After demonstrating that Tls can control flagellar gene expression and motility in MG1655, as it does in S13, we decided to perform further characterization of the Tls function in the MG1655 background. This is now explicitly stated in the text (lines 204-205). We see it as an advantage for functional studies that MG1655 does not have a native tls gene, which can rule out that the observed effects require some S13-specific factors. Moreover, this allowed us to perform all microscopy work at the lowest (S1) biosafety level, which would not be possible with the S2-level strain S13. Although we agree that studying further details of the Tls function in the S13 background, including its possible regulation by other S13-specific cellular factors, is interesting, for self-consistency we would prefer to pursue it in detail in our future work rather than replicating in S13 some of the experiments that were already performed in MG1655.

- The authors show that Tls switches its localization upon transfer from liquid to solid substrate or vice versa within 30-60 hours. It would be interesting to address if gene expression and motility follow a similar timing. Such experiments would help strengthening the connection between Tsl localization, gene expression and cellular behavior.

We have not performed these experiments because we would expect that changes in the activity of fluorescent reporter and in motility to be substantially slower than changes in the Tls localization, given the delays associated with the maturation time of GFP and its stability on one hand, and slow biogenesis of flagellar apparatus and its stability on the other hand. These delays would be there regardless of the underlying mechanism or the connection to Tls localization, meaning that these additional measurements might help us to understand the underlying mechanism.

- The authors original observations are that a tsl mutant shows a higher fraction of motile cells and that Tsl localization is influenced by the surface/liquid environment. While these are single cell observations, the authors do not follow up the link to gene expression at the single cell level. Scoring Tsl (or TlsN-mCherry) localization and gene expression in the same cells under different conditions would provide a powerful experimental link between individual cell behavior and FlhDC-mediated transcriptional control.

Although we agree that this could be an elegant experiment, one problem here is that the GFP levels reflect *fliC* reporter activity over several cell generations (because GFP is stable) and not only in this particular cell and at this particular time point. Thus, proper correlation with the Tls localization would require time-lapse imaging of both localization and reporter activity in growing cell lines, e.g. in a microfluidic device, which is not an easy experiment to do. Moreover, although our new experiments do show recruitment of FlhDC to the foci formed by Tls and confirm the proximity between these proteins by FRET, we assume that also smaller Tls complexes that are not visible as large foci can sequester FlhDC and the formation of large foci might not be necessary for the underlying regulation. We now explain better in the Discussion (lines 371-376) that it is primarily the sequestration of FlhDC away from DNA by the Tls-FlhDC complex and not only localization of this complex to the pole that we believe is required for the observed regulation.

- For instance, the idea that *E. coli* induces its *fla* and *che* genes specifically on surfaces is counterintuitive as one would rather expect downregulation of motility genes during surface adaptation. Can the authors provide some model or rationale for the reader to be able to grasp the physiological context or the general idea behind their observations? In general, I find the discussion rather uninspiring and superficial. E.g., it would be highly interesting for the reader if the authors would be discussing the FlhDC sequestration model in more detail and deliberate on how it could be further scrutinized.

We now generally improved the Discussion to better explain our hypothesis how the underlying regulation through sequestration of FlhDC by Tls may work (lines 371-408) and how it is related to the physiology of *E. coli* (lines 416-432). Regarding the specific point raised by the Reviewer: One interpretation (also following the argument in our previous work, Laganenka et al., 2020) is that the observed regulation is important to activate motility in porous media encountered by *E. coli* in our GI or urinary tract, such as mucus, which is mimicked in our experiments by soft (0.27%) agar.

- The authors show that FlhDC localizes to the cell poles in a Tsl-dependent manner (Fig. 7). However, polar foci are very weak, FlhDC levels seem to fluctuate quite a bit from cell to cell and co-localization experiments that could help clarify what is going on and strengthen a direct role of Tsl in FlhDC recruitment are missing. Furthermore, these experiments were carried out in the *E. coli* lab strain and should ideally be repeated in *E. coli* S13.

As suggested by the Reviewer, we have now performed localization and FRET experiments, which confirmed that the two proteins are parts of the same complex in liquid-grown cells, but not when cells are growing on solid medium. These data are now shown in Figure 6F,G and Expanded View Figure 4 and discussed in the text (lines 305-319). As argued above, since Tls is functional in MG1655 we decided to consistently use this background for further characterization of cellular function of Tls.

Minor points:

- To better illustrate the truncation of Tsl it would be instructive to show the full-length structure of Trg in Fig. 1B with its periplasmic, membrane-spanning and cytoplasmic parts clearly marked.

We fully agree with the Reviewer and changed the presentation of Trg structure accordingly. Although we assume that Tls can also form a dimer, in the absence of direct experimental evidence we decided to leave its predicted structure as a monomer.

- Please add a simple control experiment excluding that plasmid-based expression of Tsl (Fig. 2E) affects E. coli growth.

We thank the Review for this comment. We now show data confirming that Tls expression has no negative effect on growth (Appendix Figure 3F,G) and mention this in the text (lines 142-144).

- p.6: 'Interestingly, the abundance of FlhD and FlhC even increased in this latter case (Fig. 3D).' Was this unexpected? Please comment.

Yes, this was unexpected, since the levels of all other flagellar proteins were downregulated. We now explain this better. This stabilization supports our model that Tls might directly interact with FlhDC, to inhibit its activity but at the same time to protect it against proteolysis (as, for example, known for the downstream regulation of FliA by FlgM in the flagellar regulatory cascade). We now mention this in the Results (lines 167-169 and lines 276-282).

- Can the authors comment on the non-flagellar or chemotaxis proteins that are up- or down-regulated in response to Tsl availability (metabolism, Type 1 pili...)? What could be a possible link to FlhDC?

As was already briefly mentioned in the text, we believe that these changes are an indirect consequence of the Tls-induced changes in FlhDC activity. Although further characterization would go beyond the scope of this manuscript, we now elaborate on this observation somewhat more extensively (lines 170-180).

- The first two paragraphs of the discussion are more like an introduction and could be moved to the intro section.

We modified the Discussion section to shorten these paragraphs and move parts of this text to the Introduction (lines 359-364 and 60-73).

Referee #3:

This work examines a unique chemoreceptor-like protein, Tls, which is widely present in the B2 phylogroup of Escherichia coli. The authors found that this protein controls bacterial motility by repressing flagellar gene expression through recruitment of FlhDC to polar foci, sequestering it away from DNA. Notably, the polar localization of Tls and its mediated inhibition of flagellar gene expression only occur during growth in liquid medium, not in porous medium or on surfaces. The N-terminal fragment of Tls retains similar functionality as the intact protein when fused to mCherry. The authors demonstrated the effect of Tls on motility regulation in both bladder and gut during infection.

Overall, Lee et al. showed that the Tls protein represses flagellar gene expression, with this repression closely linked to both Tls polar foci formation and the cellular growth environment. They also demonstrated that the TlsN-mCherry fusion functions similarly to intact Tls, showing both gene repression activity and polar localization. While most results are convincing, the mechanism and physiological significance require further explanation.

We thank the Reviewer for summarizing our key findings and finding most of our results convincing, and also for helpful suggestions for additional experiments. Although full understanding of the underlying molecular mechanisms clearly requires future investigations, we believe that additional experiments suggested by this and other referees helped to further support and sharpen our model.

Here are specific points:

1. The authors' conclusion about TIs not affecting chemotaxis, based on comparing S13 and S13 Δ TIs motility on TBSA, needs additional validation. Multiple factors influence TBSA assays, including growth, motility, and chemotaxis. Additional controls are needed, such as chemotaxis-defective strains, or direct chemotaxis assays using ibidi channels.

We rephrased the corresponding sentence in the Results to say that TIs has apparently no major effect on either motility or chemotaxis in soft agar (lines 104-109). We have further performed additional measurements along the line suggested by the Reviewer, showing that expression of TIs in MG1655 partly inhibits its motility but not chemotactic bias (Expanded View Figure 1D,E) and mention this in the text (lines 144-149).

2. The authors report improved motility of S13 Δ TIs cells in liquid TB medium, with more swimmers but unchanged swimming speed and reorientation frequency. Two issues arise: First, the trajectories in Figure 2B show typical surface-associated circular swimming, making it unclear how reorientation frequency was determined. Second, for surface-located trajectories, it's important to distinguish whether non-motile bacteria are surface-attached or undergoing Brownian motion. Surface attachment would suggest TIs affects adhesion protein secretion, while Brownian motion would indicate effects on motor or flagellar filament synthesis.

We have now performed a more detailed analysis of the effects of TIs on motility, in S13 as well as in MG1655 strains. These data are now shown in Figure 2C-E and Appendix Figure 3B and Expanded View Figure 1B,C and discussed in the text (lines 115-120 and lines 138-139). They revealed that the effect of TIs on the swimming fraction is somewhat larger than suggested by our initial measurements, and also the effects on the swimming speed and on tumbling are significant although modest. Although the Reviewer is right and the tracks shown in Figure 2B were acquired at the surface (we now specifically state it in the legend (lines 992-994)), the analysis of motility was performed away from the surface and thus not affected by potential cell interactions with the surface. This is now also mentioned in the Methods section and in the legend of Figure 2 (lines 463-512 and lines 996-998).

3. The study should include motility measurements in tethering buffer, which is standard for motility and chemotaxis research.

We sincerely apologize for not explaining the motility assay well enough. The measurements were indeed performed for bacteria diluted in motility medium (nearly the same as tethering buffer). We now describe the motility measurements in greater detail in the Methods (lines 463-512).

4. The reported TIs-mediated downregulation of flagellar and chemotaxis proteins in S13 and MG1655 seems inconsistent with minimal effects on swimming speeds and frequency of cell reorientations in liquid. This discrepancy requires explanation.

As mentioned above, we have now performed a more thorough characterization of effects on TIs on motility, not only in S13 but also in MG1655. These new experiments showed that the effect of TIs on the swimming fraction is somewhat larger than suggested by our initial measurements, and also the effects on the swimming speed and on tumbling are significant although modest. In general, effects of TIs on motility are consistent with changes in gene expression but, as mentioned by the Reviewer, they are milder. We hypothesize that this might be explained by the saturation of motility at high gene expression, as shown in our recent publication that quantitatively relates activity of *fliC* reporter with motility (Lisevich et al, 2025). This means that motility may be less sensitive to the inhibition by TIs that changes in expression, particularly at high expression levels. We discuss this in the text (lines 139-142).

5. The fourth and sixth Results sections both address N-terminal region function and probably should be combined. While TIsN-mCherry functions similarly to TIs in both S13 and MG1655 strains by repressing the flagellar genes, using S13 for localization studies would be more appropriate given its natural expression of TIs. Additional questions include whether TIsN functions without fusion partners and whether the E2348/69 TIs variant shows similar activity.

We agree and have now rearranged the figures and the text accordingly (Figures 4-7 and lines 182-356). We rather see it as an advantage for functional studies that MG1655 does not have native *tIs* gene, since we can rule out that the observed effects are due to some S13-specific factors. Thus, after demonstrating that TIs can control flagellar gene expression and motility in MG1655, we decided to consistently perform further characterization of the TIs function in the MG1655 background. Moreover, this allowed us to perform all microscopy work at the lowest (S1) biosafety level, which would not be possible with the S2-level strain S13. As requested by the Reviewer, we further tested the localization of the TIsN version present in the mentioned *E. coli* isolates E2348/69. Consistent with it lacking the critical arginine residue, this version of TIsN-mCherry does not localize (Figure 5A,B) and lines 249-251 in the text, further confirming the specificity of our observations.

6. While the authors show that surface-dependent localization and flagellar gene repression do not require the chemotaxis signaling system, more discussion is needed on how TIs or TIsN aggregate at cell poles during liquid growth. The authors mention biomolecular condensates. Even if that was the case, why they appear at cell poles?

Biomolecular condensates (as well as other large protein aggregates) are commonly found at the bacterial cell poles, likely due to the exclusion from the central region containing bacterial chromosome as well as to slower protein diffusion at cell poles. We now mention this explicitly in the Discussion and cite corresponding literature (lines 373-375).

7. The FlhDC recruitment analysis in MG1655 reveals weak polar aggregation of non-mutated FlhDC, with enhanced aggregation after mutation. This may reflect insufficient recruitment capacity of non-native TIs to overcome DNA binding. Using the S13 strain would provide more convincing results. Additionally, colocalization studies of TIs and FlhDC could clarify their interaction at cell poles.

As mentioned above, we rather see it as an advantage for functional studies that MG1655 does not have native *tIs* gene, since this can rule out that the observed effects are due to some unidentified S13-specific factors. We believe that incomplete recruitment of FlhDC-GFP to the TIs foci may be rather due to relatively high expression levels of FlhDC fusion compared to the normally very low native level of FlhDC expression or to the impact of GFP

fusion on the ability of TIs to sequester FlhDC. We now clarify in the text that the aim of this experiment was to demonstrate that TIs can sequester FlhDC to the foci it forms, and not to provide a direct quantitative comparison between the extent of recruitment and that of gene expression (lines 288-294). Although we fully agree that such detailed characterization would be very interesting, it might require single-molecule studies for the chromosomally integrated fusion which would go beyond the scope of this manuscript. We now expanded the discussion of our results and their interpretation in the Discussion section (lines 371-379). However, we did perform the study of co-localization and FRET between TIs and FlhDC fusion, confirming our model that TIs can recruit FlhDC (now shown in Figure 6F,G and Expanded View Figure 4). We thank the Reviewer for this suggestion.

Dear Victor,

Thank you for submitting a revised version of your manuscript. It has now been seen by all original reviewers, and I have copied their comments below.

While reviewers #1 and #3 are generally satisfied with the revision, reviewer #2 finds that several of their initial requests were not clarified satisfactorily. In addition, reviewer #1 also has a few minor requests. Please address these points by both reviewers via extended discussion and textual clarifications.

Additionally, there are a few editorial points that need addressing before I can extend official acceptance of the manuscript:

1. Please ensure that the manuscript section order is as follows: Title page - Abstract - Keywords - Introduction - Results - Discussion - Methods - Data Availability - Acknowledgements - Disclosure and Competing Interests Statement - References - Figure Legends - Table(s) - Expanded View Figure Legends.
2. CRedit has replaced the traditional author contributions section because it offers a systematic, machine-readable author contributions format that allows for more effective research assessment. Please remove the Authors Contributions from the manuscript and use the free text boxes beneath each contributing author's name in our online submission system to add specific details on the author's contribution. More information is available in our guide to authors.
3. All Materials and Methods need to be described in the main text using our 'Structured Methods' format. According to this format, the Methods section includes a Reagents and Tools Table (listing key reagents, experimental models, software and relevant equipment and including their sources and relevant identifiers) followed by a Methods and Protocols section describing the methods, ideally using a step-by-step protocol format. The aim is to facilitate adoption of the methodologies across labs. Please download and fill our Reagents and Tools Table template (.docx), which you can find in our author guidelines: <https://www.embopress.org/page/journal/14602075/authorguide#structuredmethods> When submitting your revised manuscript, please do not include the Reagents and Tools Table in the Methods section of the manuscript but upload it as a separate file choosing the file type "Reagent Table". An example of a Method paper with Structured Methods can be found here: <https://www.embopress.org/doi/10.15252/msb.20178071>.
4. There is a reference to "data not shown" on page 13. According to our policy, which does not permit references to "data not shown", please include this information in the Appendix. Please see also <https://www.embopress.org/page/journal/14602075/authorguide#unpublisheddata>.
5. In the Appendix, please add a front page with the title "Appendix for + manuscript tile" and a brief table of contents with page numbers for the listed items.
6. For the main figures, please upload source data files as one zip folder per figure that includes both images and numerical data. E.g., all the source data files for figure 1 need to be saved in a single folder and this needs to be zipped and then uploaded as "SD figure 1.zip" file. For EV and/or appendix figures, source data can be zipped together in a single folder.
7. During our routine image checks, we noticed that the microscopy panels across the figure set appear pixelated. This can be a common result of converting original 16-bit TIFF images to a RGB or PDF format for publication. While it is not a cause for concern, it can sometimes give the impression of image alteration to critical readers. Please include higher-resolution microscopy images for the entire figure set.
8. Our data editors have flagged the following issues in figure legends that need correcting:
 - Please indicate the statistical test used for data analysis in the legends of figures 3C, D.
 - Please provide information on the number and nature of replicates in the legends of figures 2C, D, E, H
 - Please define the white arrows in the legend of figure 6A, D.
9. Papers published in The EMBO Journal are accompanied online by a 'Synopsis' to enhance discoverability of the manuscript. It consists of A) a short (1-2 sentences) summary of the findings and their significance, B) 3-4 bullet points highlighting key results and C) a synopsis image that is 550x300-600 pixels large (width x height, jpeg or png format). You can either show a model or key data in the synopsis image. Please note that the image size is rather small and that text needs to be readable at the final size.

With best wishes,

leva

We realize that it is difficult to revise to a specific deadline. In the interest of protecting the conceptual advance provided by the work, we recommend a revision within 3 months (9th Oct 2025). Please discuss the revision progress ahead of this time with the editor if you require more time to complete the revisions.

Referee #1:

This revised version of the manuscript "Control of flagellar gene expression by the receptor-like regulator in pathogenic *Escherichia coli*" by Lee and colleagues describes the regulatory role of the truncated receptor Tls in the expression of flagellar and chemotaxis genes. This time, the authors provide the molecular mechanisms underlying this regulation, which they elucidated using a quantitative cell biology approach. Specifically, the mechanism involves the relocation of Tls from the bacterial cytoplasm to the cell pole in response to mechanical stimuli (i.e., the transition from a solid surface to a liquid medium). Other factors that remain to be identified, might be involved in this localization switch. At the cell pole, Tls sequesters FlhDC away from DNA, thereby inactivating the transcription of Class 2 and 3 flagellar and chemotaxis genes. Results also suggest that this mechanism likely required the short Tls N-terminal sequence and the ability of the protein to dimerize.

In this new version, the interaction between Tls and FlhDC is quantitatively assessed using a FRET-based approach, a technique in which the Sourjik lab has strong expertise.

The addition of these convincing new results makes the manuscript suitable for publication in The EMBO Journal.

A few minor additions would further strengthen the manuscript:

Figure 1A: Beside the loss of the N-terminal region of Trg, the MokB gene is also lost. Maybe, mention in the discussion that this gene is unrelated to the Trg function and motility in general?

Figure 2A: Could the slightly reduced swarming halo of S13 tls be due to the increased tumbling frequencies observed in liquid?

Figure 2C-E: Please specify the number of analyzed cells / events. Also specify how the tumbling bias is expressed (number of tumbling events per time unit?).

Figure 2G: Please mention that UTI89 tls does not have the same growth defect as the corresponding mutant in S13.

Figure 6A and D: Please draw the cell contour (or find another way) to help the reader appreciate the relocation from DNA to the cytoplasm.

Figure S3C-D: Please specify the number of analyzed cells.

Line 248: add reference to Figure S8C-D.

It would be nice to add the Alpha fold of the dimeric Tls or TlsN-sfGFPN149Y or TlsN-sfGFPR206A.

Referee #2:

The revised manuscript shows clear improvement, primarily due to the addition of FRET data demonstrating that Tls interacts—either directly or indirectly—with FlhDC. However, several experimental approaches intended to strengthen the proposed model remain incomplete. Notably, the cell biology experiments were conducted exclusively in a laboratory strain (MG1655) that does not express a native copy of tls. Given that the model heavily relies on these observations, this reviewer strongly encourages validation of key findings in the native strain. The argument that Tls is fully functional in MG1655 is not entirely convincing. While Tls overexpression in this background strongly affects fla gene expression, its impact on motility is comparatively weak. It is possible that the Tls-dependent polar localization of FlhDC, which appears faint in MG1655, might be more pronounced and convincing in the native context. Overall, demonstrating that Tls localization and FlhDC sequestration also occur in the native strain would substantially enhance the model.

A second potential weakness concerns the causal relationship between tls and motility, which remains incompletely resolved. The motility phenotype was not consistently observed across all strains tested (e.g., absent in UTI89). In strain S13, only fla gene expression was monitored in the Tls tuning experiments, with no motility data provided. Likewise, in MG1655, full induction of tls expression nearly abolishes fliC transcription but produces only a modest reduction in motility. Most importantly, the

authors have not addressed whether the TIs-dependent effect on fla expression functionally requires FlhDC.

Finally, the role of the N-terminal region of TIs remains unclear. The data suggest that this domain is responsible for both polar localization of TIs and for recruiting FlhDC to the cell poles. The requirement for a dimeric conformation indicates that subcellular localization and/or sequestration may be driven by multivalent protein-protein interactions involving this part of the protein. While the authors briefly mention the possibility of higher-order complexes or biomolecular condensates, a more in-depth discussion of this surprising finding-and possible mechanistic explanations-would significantly strengthen the manuscript.

Referee #3:

The revision addressed most of my comments, thus its acceptance for publication is recommended.

Responses to reviewers**Referee #1:**

This revised version of the manuscript "Control of flagellar gene expression by the receptor-like regulator in pathogenic *Escherichia coli*" by Lee and colleagues describes the regulatory role of the truncated receptor Tls in the expression of flagellar and chemotaxis genes. This time, the authors provide the molecular mechanisms underlying this regulation, which they elucidated using a quantitative cell biology approach. Specifically, the mechanism involves the relocation of Tls from the bacterial cytoplasm to the cell pole in response to mechanical stimuli (i.e., the transition from a solid surface to a liquid medium). Other factors that remain to be identified, might be involved in this localization switch. At the cell pole, Tls sequesters FlhDC away from DNA, thereby inactivating the transcription of Class 2 and 3 flagellar and chemotaxis genes. Results also suggest that this mechanism likely required the short Tls N-terminal sequence and the ability of the protein to dimerize.

In this new version, the interaction between Tls and FlhDC is quantitatively assessed using a FRET-based approach, a technique in which the Sourjik lab has strong expertise.

The addition of these convincing new results makes the manuscript suitable for publication in The EMBO Journal.

We thank the Reviewer for this positive re-evaluation of our manuscript and recommending it for publication, and for her/his very helpful previous feedback and the remaining suggestions for improvement.

A few minor additions would further strengthen the manuscript:

Figure 1A: Beside the loss of the N-terminal region of Trg, the MokB gene is also lost. Maybe, mention in the discussion that this gene is unrelated to the Trg function and motility in general?

We agree that it is worthwhile mentioning and did it in the legend of Figure 1 (lines 1023-1025).

Figure 2A: Could the slightly reduced swarming halo of S13 Δ Tls be due to the increased tumbling frequencies observed in liquid?

This is indeed possible, and we now mention it in the text (lines 120-122), along with our previous (and still preferred) explanation relating it to the minor difference in the growth rate. We thank the Reviewer for pointing out this possible alternative.

Figure 2C-E: Please specify the number of analyzed cells / events. Also specify how the tumbling bias is expressed (number of tumbling events per time unit?).

Tumbling bias was defined in the Methods (lines 511), as the fraction of time spent tumbling (and does not have a unit). We further adjusted the phrasing in the text (lines 118-120) and in the figure legend (lines 1048-1051) to make it clearer.

The data in Fig 2C,D were measured by DDM so individual cells are not identified, but there should be an order of 10^2 - 10^3 cells in the field of view at any given time, with many going in and out of the field of view during the 100 s recording. This is now better explained in the Methods (lines 496-500).

The data in Fig 2E were indeed measured by particle tracking, and we now specify the total number of analyzed trajectories (over all 3 replicates) for each condition in the figure legend (lines 1048-1051).

Figure 2G: Please mention that UTI89 Δ Tls does not have the same growth defect as the corresponding mutant in S13.

We modified the corresponding sentence in the Results to mention this (lines 126-128).

Figure 6A and D: Please draw the cell contour (or find another way) to help the reader appreciate the relocation from DNA to the cytoplasm.

We thank the Reviewer for pointing out this point. We now enlarged the images for better visibility and also removed the arrows.

Figure S3C-D: Please specify the number of analyzed cells.

As explained in our response to a similar question regarding Figure 2C,D above, these data were measured by DDM so individual cells were not identified, but there should be an order of 10^2 - 10^3 cells in the field of view at any given time, with many going in and out of the field of view during the 100 s recording. This is now better explained in the Methods (lines 496-500).

Line 248: add reference to Figure S8C-D.

We now added reference to this figure, as suggested.

It would be nice to add the Alpha fold of the dimeric Tls or TlsN-sfGFPN149Y or TlsN-sfGFPR206A.

We now show the prediction of the dimeric structure for Tls in Figure 1B.

Referee #2:

The revised manuscript shows clear improvement, primarily due to the addition of FRET data demonstrating that Tls interacts-either directly or indirectly-with FlhDC. However, several experimental approaches intended to strengthen the proposed model remain incomplete. Notably, the cell biology experiments were conducted exclusively in a laboratory strain (MG1655) that does not express a native copy of *tls*. Given that the model heavily relies on these observations, this reviewer strongly encourages validation of key findings in the native strain. The argument that Tls is fully functional in MG1655 is not entirely convincing. While Tls overexpression in this background strongly affects *fla* gene expression, its impact on motility is comparatively weak. It is possible that the Tls-dependent polar localization of FlhDC, which appears faint in MG1655, might be more pronounced and convincing in the native context. Overall, demonstrating that Tls localization and FlhDC sequestration also occur in the native strain would substantially enhance the model.

We thank the Reviewer for acknowledging that the manuscript has clearly improved and for suggesting several additional points that could further strengthen it. We have now performed FRET and colocalization experiments in the S13 background as suggested by the Reviewer (now shown in Figure EV4C-E). These confirmed that the interaction between Tls and FlhDC is also observed in the S13

background, and that this sequestration depends on growth conditions (liquid vs surface) in the same way as in the MG1655 background. These experiments had to be performed in a very time-consuming way on a different microscope (because of the biosafety-related limitation in the microscope usage), where FRET had to be measured in individual cells using pixel-by-pixel scanning instead of measurements of the integrated intensity for the entire population. This was one of the reasons why we preferred to use MG1655 as a standard background strain. But it allowed us to separately quantify FRET in cells that either show visible foci or not. The results supported our hypothesis that significant sequestration of FlhDC by Tls already occurs in cells without visible foci (presumably in small cytoplasmic complexes that are not yet visible as distinct foci in a regular epifluorescence microscope).

A second potential weakness concerns the causal relationship between Tls and motility, which remains incompletely resolved. The motility phenotype was not consistently observed across all strains tested (e.g., absent in UT189). In strain S13, only *fla* gene expression was monitored in the Tls tuning experiments, with no motility data provided. Likewise, in MG1655, full induction of Tls expression nearly abolishes *fliC* transcription but produces only a modest reduction in motility. Most importantly, the authors have not addressed whether the Tls-dependent effect on *fla* expression functionally requires FlhDC.

We have added the requested experiment showing that Tls tuning causes similar gradual inhibition of swimming velocity in the S13 Δ *tls* background (now shown in Figure S3E), which is similar to already shown inhibition in MG1655. Regarding the other point about the dependence of the Tls-regulatory effect on FlhDC: We do believe that our observation of the in-vivo interaction between Tls and FlhDC, and of Tls dependent relocalization of FlhDC, as well as the fact that expression of all FlhDC-dependent genes (but not of FlhDC itself) changes depending on Tls, strongly supporting our explanation that Tls post-translationally regulates FlhDC activity through its sequestration. We do not see any other way to test the FlhDC-dependence of the Tls effect (at least not in vivo), since FlhDC is required for the expression of all other flagellar genes.

Finally, the role of the N-terminal region of Tls remains unclear. The data suggest that this domain is responsible for both polar localization of Tls and for recruiting FlhDC to the cell poles. The requirement for a dimeric conformation indicates that subcellular localization and/or sequestration may be driven by multivalent protein-protein interactions involving this part of the protein. While the authors briefly mention the possibility of higher-order complexes or biomolecular condensates, a more in-depth discussion of this surprising finding-and possible mechanistic explanations-would significantly strengthen the manuscript.

We fully agree with the Reviewer that the function of the N-terminal region of Tls is highly interesting. But we already have an entire paragraph in the Discussion (lines 419-430) hypothesizing about possible mechanism, and we feel that more extensive speculations are not justified at this point.

Referee #3:

The revision addressed most of my comments, thus its acceptance for publication is recommended.

We thank the Reviewer for this positive re-evaluation of our manuscript and recommending it for publication.

Dear Victor,

Thank you for addressing the final editorial points. I apologise for the slow process from our side due to the high number of submissions that we receive at the moment. I am now pleased to inform you that your manuscript has been accepted for publication - congratulations!

Before we forward your manuscript to our publishers, I would like to propose some minor edits in the manuscript title, abstract and synopsis (please see below and in the attached file). I have also written a short blurb that will accompany the title of your manuscript in our online table of contents. Please take a look and let me know if any corrections or adjustments are needed.

Title:

Control of flagellar gene expression by a chemotaxis receptor-like regulator in pathogenic *Escherichia coli*

Blurb:

Chemoreceptor-like protein Tls represses bacterial motility in a substrate-dependent manner in a subgroup of *E. coli* including extraintestinal pathogenic strains.

Synopsis:

Chemotaxis receptors typically mediate biased movement of motile bacteria in environmental gradients by controlling rotation of the flagellar motor. This study identifies a chemotaxis receptor in a model bacterial pathogen that instead controls expression of motility genes.

- B2 phylogroup of *Escherichia coli* possesses a novel motility regulator, Tls, that evolved from the chemotaxis receptor Trg.
- Instead of signalling to flagellar motor through the chemotaxis pathway, Tls controls flagellar gene expression.
- Tls sequesters FlhDC, the master transcriptional activator of flagellar genes, depending on the mechanical properties of the growth environment.
- Tls affects pathogen interaction with host cells and proliferation in the mouse model.

If you have any questions, please do not hesitate to contact the Editorial Office. Thank you for this contribution to The EMBO Journal and congratulations on a nice study!

With best wishes,

Ieva

Ieva Gailite, PhD
Senior Scientific Editor
The EMBO Journal
Meyerohofstrasse 1
D-69117 Heidelberg
Tel: +4962218891309
i.gailite@embojournal.org

>>> Please note that it is The EMBO Journal policy for the transcript of the editorial process (containing referee reports and your response letter) to be published as an online supplement to each paper. If you do NOT want this, you will need to inform the

Editorial Office via email immediately. More information is available here: https://www.embopress.org/transparent-process#Review_Process